# Establishment of H3K9me3-dependent heterochromatin during embryogenesis in *Drosophila miranda*

Kevin H-C Wei, Carolus Chan, Doris Bachtrog*

Department of Integrative Biology, University of California, Berkeley, Berkeley, United States

**Abstract** Heterochromatin is a key architectural feature of eukaryotic genomes crucial for silencing of repetitive elements. During *Drosophila* embryonic cellularization, heterochromatin rapidly appears over repetitive sequences, but the molecular details of how heterochromatin is established are poorly understood. Here, we map the genome-wide distribution of H3K9me3-dependent heterochromatin in individual embryos of *Drosophila miranda* at precisely staged developmental time points. We find that canonical H3K9me3 enrichment is established prior to cellularization and matures into stable and broad heterochromatin domains through development. Intriguingly, initial nucleation sites of H3K9me3 enrichment appear as early as embryonic stage 3 over transposable elements (TEs) and progressively broaden, consistent with spreading to neighboring nucleosomes. The earliest nucleation sites are limited to specific regions of a small number of recently active retrotransposon families and often appear over promoter and 5' regions of LTR retrotransposons, while late nucleation sites develop broadly across the entirety of most TEs. Interestingly, early nucleating TEs are strongly associated with abundant maternal piRNAs and show early zygotic transcription. These results support a model of piRNA-associated co-transcriptional silencing while also suggesting additional mechanisms for site-restricted H3K9me3 nucleation at TEs in pre-cellular *Drosophila* embryos.

*For correspondence:
dbachtrog@berkeley.edu

**Competing interests:** The authors declare that no competing interests exist.

## Introduction

The separation of eukaryotic genomes into transcriptionally active euchromatin and silenced heterochromatin is a fundamental aspect of eukaryotic genomes (*Allshire and Madhani, 2018*). Heterochromatin is the gene-poor, transposon-rich, late-replicating, and tightly packaged chromatin compartment that was first cytologically defined over 90 years ago (*Heitz, 1928*), in contrast to euchromatin, the gene-rich, lightly packed form of chromatin. These domains show characteristic distributions across eukaryotic genomes and are distinguished by unique sets of histone modifications (*Peng and Karpen, 2008*; *Elgin and Reuter, 2013*). Heterochromatin is found predominantly at repetitive sequences, which mainly correspond to pericentromeres, the dot and the Y chromosome in flies, and is marked by tri-methylation of histone H3 lysine 9 (H3K9me3) (*Elgin and Reuter, 2013*).

Heterochromatin formation, and the boundary between heterochromatic and euchromatic domains is established during early development. In *Drosophila melanogaster*, constitutive heterochromatin is not observed cytologically in the initial zygote, but emerges during blastoderm formation (*Vlassova et al., 1991*; *Lu et al., 1998*) (developmental stage 4; see *Figure 1A,B*). Chromatin assembly during this period is prior to any widespread zygotic transcription and dependent on maternally loaded RNA and proteins (*Elgin and Reuter, 2013*). Analysis of an inducible reporter gene has found that silencing occurs at the onset of gastrulation (end of stage 6), about 1 hr after heterochromatin is visible cytologically (*Lu et al., 1998*). The extent of silencing increases as

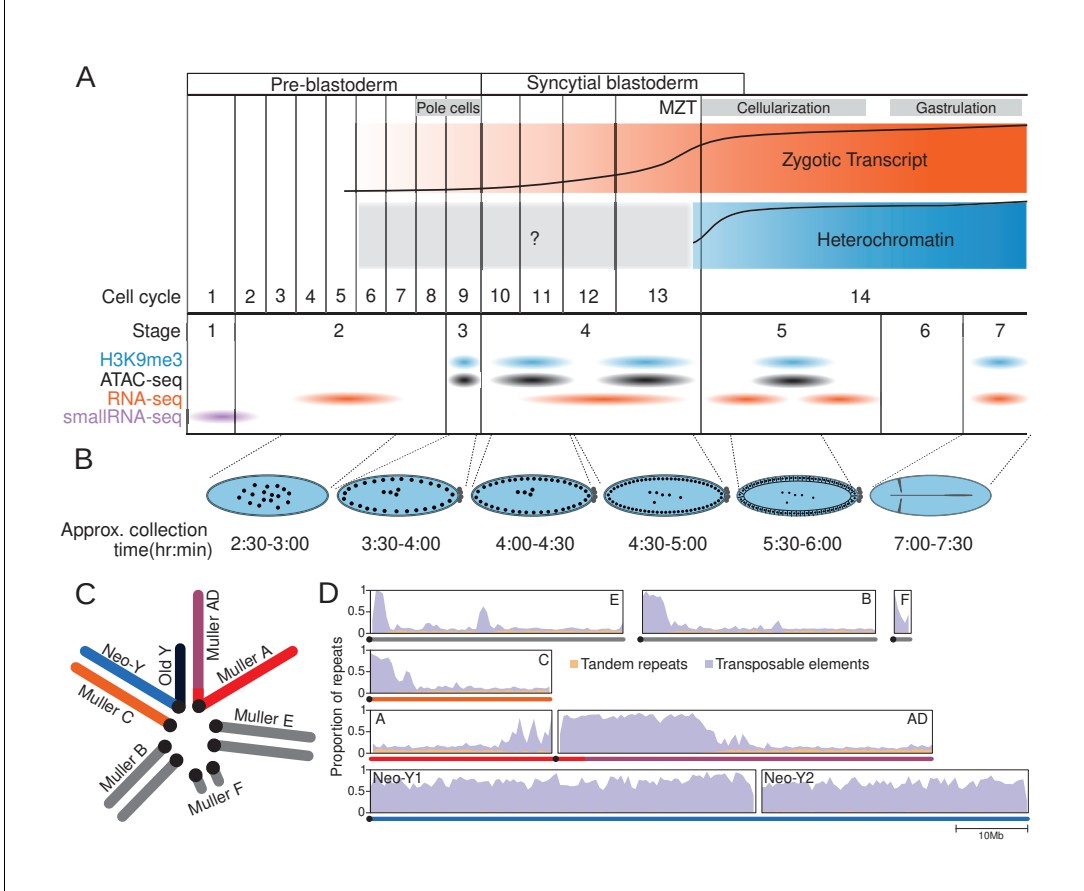

**Figure 1.** The repeat-rich genome of *Drosophila miranda* as a model to characterize heterochromatin establishment. (**A**) Time course of embryonic development. Major landmarks including the maternal to zygotic transition (MZT) are labeled. The orange and blue track depicts the approximate amount of zygotic expression and heterochromatin, respectively, throughout development. The cell cycle numbers and their corresponding embryonic stages are labeled. (**B**) Cartoon diagram of the embryonic landmarks used for staging. The approximate time points of embryo collection are labeled under the embryo diagrams. (**C**) Karyotype of *D. miranda* male. Muller elements are labeled and the sex chromosomes are color-coded: neo-Y (blue), ancestral Y (dark navy), neo-X (orange), and X (red and purple). (**D**) Repeat content of the *D. miranda* genome assembly. The cumulative repeat content for each chromosome is depicted, with tandem repeats in orange and transposable elements in purple; unless otherwise stated, these repeat classes will be distinguished by these colors throughout the manuscript. Karyotypes are drawn below the graphs with the chromosomes color-coded as in (**C**). The online version of this article includes the following figure supplement(s) for figure 1:

**Figure supplement 1.** Correlation between H3K9me2 and H3K9me3 enrichment.

**Figure supplement 2.** Quantile-informed spike-in normalization procedure.

embryonic development progresses, and by stage 15 (dorsal closure, between 11.5 and 13 hr of development), silencing patterns reminiscent of those for third-instar larvae are established (*Lu et al., 1998*).

The role of H3K9me3-associated heterochromatin in genome regulation and silencing of repetitive elements is well established. H3K9me3 is deposited by a conserved class of histone methyltransferases (*Rea et al., 2000*; *Nakayama et al., 2001*) and provides a high-affinity binding site for HP1 proteins (*Bannister et al., 2001*; *Lachner et al., 2001*). HP1 proteins can oligomerize, resulting in local chromatin compaction (*Hiragami-Hamada et al., 2016*), and help establish heterochromatin domains possibly via phase separation (*Larson et al., 2017*; *Strom et al., 2017*, but see *Erdel et al., 2020*). HP1 proteins can also recruit additional histone methyltransferases, thereby enabling spreading of heterochromatin (*Canzio et al., 2013*). Thus, H3K9me3 deposition is a crucial step in the formation of heterochromatin and the suppression of repetitive sequences, but the molecular mechanisms of H3K9me3 targeting and recruitment are not fully understood.

Several studies have suggested that small RNA-mediated silencing pathways can initiate the formation of heterochromatin (*Holoch and Moazed, 2015*; *Allshire and Madhani, 2018*). This was first identified in fission yeast, where mutations in components of the RNAi pathway showed disrupted heterochromatin silencing in centromeres (*Volpe et al., 2002*; *Hall et al., 2002*). Further studies revealed that siRNAs with sequence homology to repeats bind to their nascent transcripts to direct the recruitment of chromatin-associated silencing factors (*Kato, 2005*; *Djupedal et al., 2005*; *Bühler et al., 2006*). In flies and mammals, a different class of small RNAs, PIWI-associated small RNAs (piRNAs; 23-29nt in *Drosophila*), are involved in the silencing of transposable elements (TEs) (*Brennecke et al., 2007*; *Aravin et al., 2007*; *Olovnikov et al., 2012*; *Czech et al., 2018*; *Ninova et al., 2019*). piRNAs are derived from TE sequences and provide heritable libraries that target TE transcripts for degradation (*Brennecke et al., 2007*; *Czech et al., 2018*).

Independent of the post-transcriptional cleavage activity (*Wang and Elgin, 2011*; *Darricarrère et al., 2013*), the piRNA pathway is required for transcriptional silencing of TE insertions in the ovary. In flies, loss of PIWI causes the loss of H3K9me3-mediated heterochromatic repression (*Klenov et al., 2011*; *Wang and Elgin, 2011*; *Sienski et al., 2012*; *Darricarrère et al., 2013*; *Klenov et al., 2014*). Similar to yeast, piRNA-guided heterochromatin formation requires transcription of the target locus (co-transcriptional silencing); Piwi–piRNA complexes bind to complementary, nascent TE transcripts and induce heterochromatin formation by recruiting heterochromatin factors, including histone methyltransferases for H3K9me3 deposition (*Sienski et al., 2012*; *Wang and Elgin, 2011*; *Le Thomas et al., 2013*; *Yu et al., 2015*; *Batki et al., 2019*).

In the developing embryo, maternally deposited PIWI is similarly required for the formation of H3K9me3 at the maternal-zygotic transition during late blastoderm (*Gu and Elgin, 2013*). However, the process and precise mechanism of rapid heterochromatin establishment across a large fraction of the genome such as the pericentromere and the Y chromosome remains poorly understood. Early cytological studies showed that heterochromatin becomes visible during blastoderm (*Vlassova et al., 1991*; *Lu et al., 1998*). More recently, in vivo immunofluorescence staining on the developing *Drosophila* embryo has shown that heterochromatin appears as early as cell cycle 13 (late stage 4) and rapidly increases through blastulation (cell cycle 14, stage 5) and requires the histone methyltransferase SetDB1 (*Yuan and O'Farrell, 2016*; *Seller et al., 2019*). However, such imaging approaches lack resolution at genome-scale and are necessarily limited by the concentration and density of the targeted sequence and proteins. Emerging heterochromatin, therefore, may not yield strong enough signal for fluorescent detection. Hence, highly sensitive genome-wide analyses of H3K9me3-dependent heterochromatin dynamics in early embryos are needed.

Here, we investigate the establishment of heterochromatin during early embryogenesis in *Drosophila miranda.* While this species lacks the genetic tools available in *D. melanogaster*, several features make *D. miranda* ideal for studying repetitive sequences and heterochromatin on a genome scale. *D. miranda* has a large repeat-rich neo-Y chromosome, which formed through the fusion of an autosome (Muller-C) with the ancestral Y around 1.5 million years ago and has since more than doubled in size due to the accumulation of TEs. The high-quality genome assembly has near end-to-end assemblies of most chromosome arms that include large fractions of heterochromatin (*Figure 1C–D*; *Mahajan et al., 2018*): roughly 30 Mb of pericentromeric heterochromatin at the X and autosomes, and over 100 Mb of the repeat-rich Y chromosome have been assembled. In contrast, the most repeat-rich genome assembly of *D. melanogaster* contains only 14.6 Mb of Y-linked sequence scattered across 106 contigs (*Chang and Larracuente, 2019*). *D. miranda*'s neo-Y chromosome is predominantly assembled into two massive contigs (53.8 and 37.2 Mb) and harbors an abundance of active single-copy genes interspersed in repeats; the mixture of genes and repeats create nonredundant junctions facilitating sensitive read mapping (*Wei et al., 2020*). Additionally, unlike *D. melanogaster* that contains Mb-sized blocks of simple satellite DNA, this species has few simple satellites in its genome (*Wei et al., 2018*), thus allowing us to map most heterochromatic regions using ChIP-seq technology. To study the formation of H3K9me3-dependend heterochromatin during early development, we adapted an ultra-low input ChIP-seq protocol using single precisely staged embryos (*Figure 1A*). Our dense sampling during early embryogenesis (genome-wide H3K9me3 profiles for >70 individual embryos) allowed us to infer spatiotemporal heterogeneity in heterochromatin establishment and evaluate the relationship between the nucleation and spreading of H3K9me3 marks to zygotic genome activation and maternally deposited piRNAs.

## Results

### Genome-wide profiling of H3K9me3 in *Drosophila* across an embryonic time course

To define the chromatin landscape before, during, and after the establishment of heterochromatin, we collected *D. miranda* (MSH22) embryos from population cages at 18°C between 210 and 420 min, to target embryonic stage 3 (before the onset of heterochromatin formation), early and late stage 4 (when some heterochromatin is first detected), stage 5 (when heterochromatin becomes cytologically visible), and stage 7 (when heterochromatin exerts its silencing effect), respectively (see *Figure 1A* and *Supplementary file 1*). Note that embryonic development is highly stereotypical across the *Drosophila* genus, with nearly identical morphological landmarks (*Kuntz and Eisen, 2014*). Embryos were examined under a light microscope to only select embryos of the correct stages, based on major morphological features to classify developmental stages (see Materials and methods, *Figure 1B*; *Lott et al., 2014*). For each stage, we collected between 8 and 20 embryos, which were subjected to single-embryo, ultra-low input ChIP-seq library preparation (*Brind'Amour et al., 2015*), with antibodies against the repressive histone mark H3K9me3. The large numbers of replicates (*Supplementary file 1*) ensure robust and sensitive profiles of H3K9me3 enrichment across the stages. We initially performed two independent IPs on individual embryos, targeting both H3K9me2 and H3K9me3 (*Figure 1—figure supplement 1*). Overall, their enrichment profiles are very similar and highly correlated (*Figure 1—figure supplement 1*). In addition, we spiked the libraries with stage 7 *D. melanogaster* embryos, so that enrichment profiles can be compared across samples and stages after spike-in normalization (see Materials and methods and *Figure 1—figure supplement 2*).

### Bulk of heterochromatin establishment occurs during early stage 4 of embryonic development

Based on cytology and microscopy, heterochromatin is first detectable at the last cell cycles in stage 4 (that is around nuclear cycle 12/13) and then becomes established rapidly during cellularization at stage 5 (*Vlassova et al., 1991*; *Lu et al., 1998*; *Yuan and O'Farrell, 2016*; *Larson et al., 2017*; *Strom et al., 2017*). However, we already see clear H3K9me3 enrichment around the pericentric regions of each chromosome at early stage 4 (that is, nuclear cycle 10/11), followed by subsequent increases in enrichment through stage 7 (*Figure 2A*). The same pattern of establishment at early stage 4 is also apparent when we inferred regions of the genome with H3K9me3 enrichment peaks; the number of H3K9me3 peaks identified sharply increases from stage 3 (n = 2202) to early stage 4 (n = 51,202), followed by proportionally smaller increases in peak numbers through the later stages (*Figure 2B*, *Figure 2—figure supplement 1*). Interestingly, as development progresses, peak sizes broaden and become less sharp (*Figure 2B,C*, *Figure 2—figure supplements 1* and *2*). After stage 3, we see a gradual increase of H3K9me3 enrichment around early peaks as embryos age (*Figure 2C*). Concordantly, regions of the genome where peaks are inferred in late stage 4, stage 5, and stage 7 already show positive H3K9me3 enrichment at early stage 4, on average (*Figure 2D*), even though they are depleted, on average, at stage 3 (*Figure 2D*). In fact, nearly 60% of the 116,791 peaks inferred for stage 7 are already enriched for H3K9me3 at early stage 4, even if the enrichment level is below the detection limit of peak calling (*Figure 2E*). Furthermore, an additional 27% of the peaks also show enrichment at stage 3 (*Figure 2E*). Similarly, for late stage 4 and stage 5, the majority of the peaks are enriched in early stage 4. Altogether, these results indicate that pericentric heterochromatin establishment in *D. miranda* likely starts at or before stage 4, followed by increases in heterochromatin levels during subsequent stages.

### Bona fide stage 3 peaks nucleate heterochromatin

While the bulk of H3K9me3 enrichment appears at early stage 4, we also identified a small number of peaks at stage 3 (n = 2202). which show sharp and narrow enrichment around their center that precipitously drops to depletion less than 100 bp away (*Figure 2B,C*). This is a strikingly different pattern from peaks called at all other stages that show elevated enrichment that span well over 1 kb away from the peak (*Figure 2B,C*). To ensure that stage 3 peaks and enrichment are not artifacts due to the small number of nuclei ('phantom peaks'; *Jain et al., 2015*), we collected ChIP-seq of

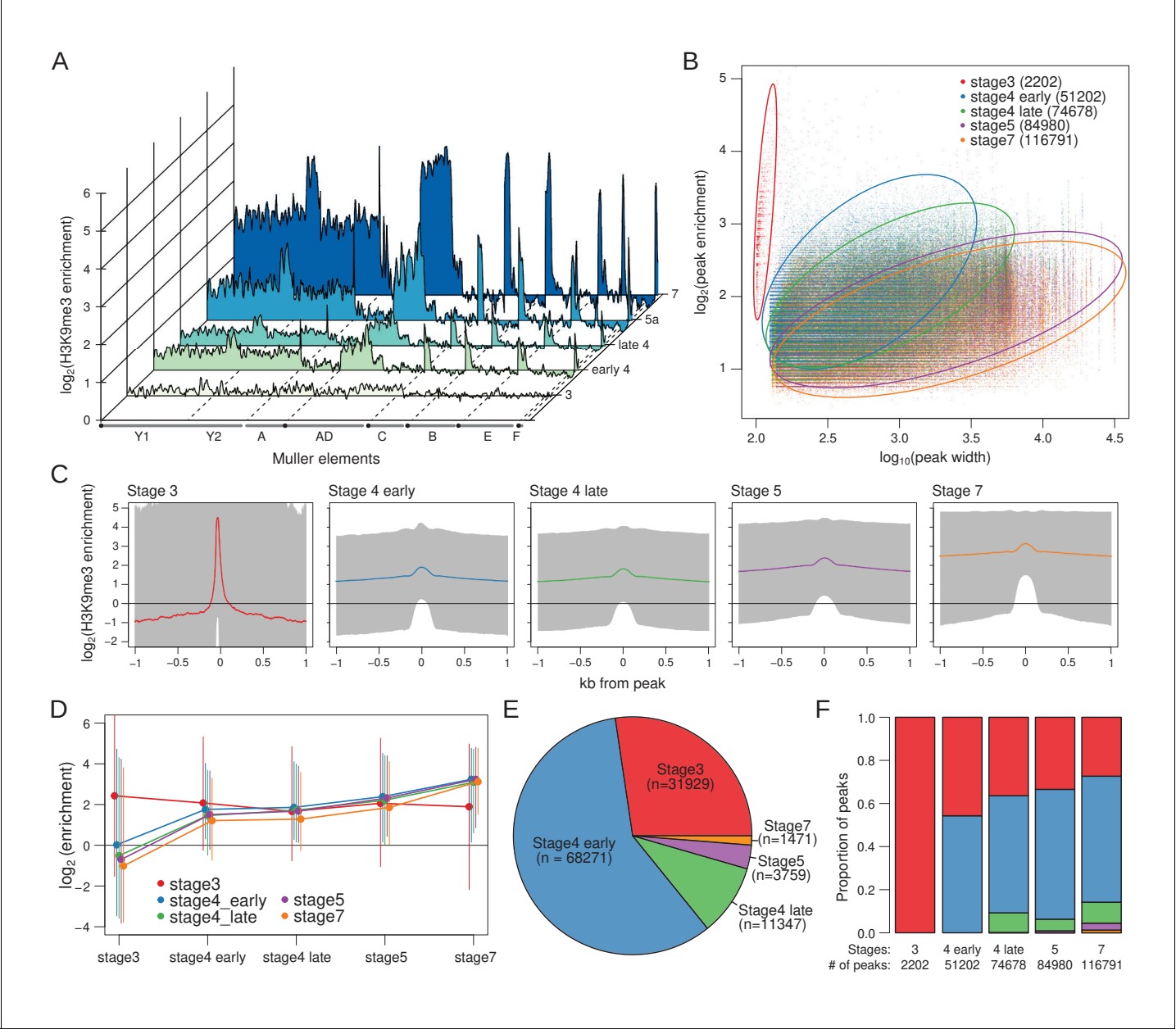

**Figure 2.** Developmental trajectory of heterochromatin enrichment and peaks. (A) Genome-wide H3K9me3 enrichment landscape through five embryonic stages. Karyotypes are depicted below the X-axis, with the centromeres marked by black circles. (B) Width and height of H3K9me3 peaks (points), as determined by MACS2, is plotted in log scale on the X- and Y-axes, respectively. Stages are color labeled with number of peaks in parentheses. Circles outline areas in which the bulk of the points of a stage reside. Unless otherwise stated, the developmental stages will be differentiated consistently with these colors henceforth. (C) Median H3K9me enrichment in log scale is plotted ±1 kb around peaks for each stage. Gray area demarcates the 95% confidence intervals. (D) H3K9me3 enrichment trajectory of peaks called for each stage across development. For every set of peaks called in each stage, the median enrichment value is plotted and connected across all developmental stages with the 95% CI demarcated by vertical lines. For example, red points and lines are the enrichment values around stage 3 peaks across all five stages. Points and CIs are horizontally staggered for clarity. (E) Colored areas in pie chart mark the proportion of stage 7 peaks that are already enriched (>1.5-fold enrichment) in previous stages. (F) Barplot format of (E), but for peaks called in every stage.

The online version of this article includes the following figure supplement(s) for figure 2:

**Figure supplement 1.** Width and height of H3K9me3 enrichment peaks across different developmental stages.

**Figure supplement 2.** Distribution of peak widths across developmental stages.

stage 3 embryos using three additional antibodies targeting H3, H3K4me3 (an active transcription mark), and H4K16ac (the dosage compensation mark). We find that many of the stage 3 heterochromatin peaks show similarly elevated enrichment in the other preparations, indicating that a subset of the peaks are indeed artifacts (*Figure 3A*). We therefore removed 1205 peaks that show enrichment in one or more of the other ChIP preparations, resulting in a set of 997 H3K9me3-specific peaks (henceforth stage 3 peaks; *Figure 3A*, *Figure 3—figure supplement 1*).

Around stage 3 peaks, H3K9me3 enrichment gradually expands across development (*Figure 3B, C*), emblematic of the spreading of the heterochromatic domain. Since a single nucleosome is wrapped by ~147 bp of DNA, the narrow width of the stage 3 peaks (median width of 125 bp) suggests that they predominantly represent single nucleosomes that nucleate the deposition of H3K9me3 in neighboring nucleosomes in subsequent stages (*Figure 3A,C*). 47.8% and 26.2% of the stage 3 peaks overlap with annotated TEs and microsatellites (*Figure 3E*). Unexpectedly, though,

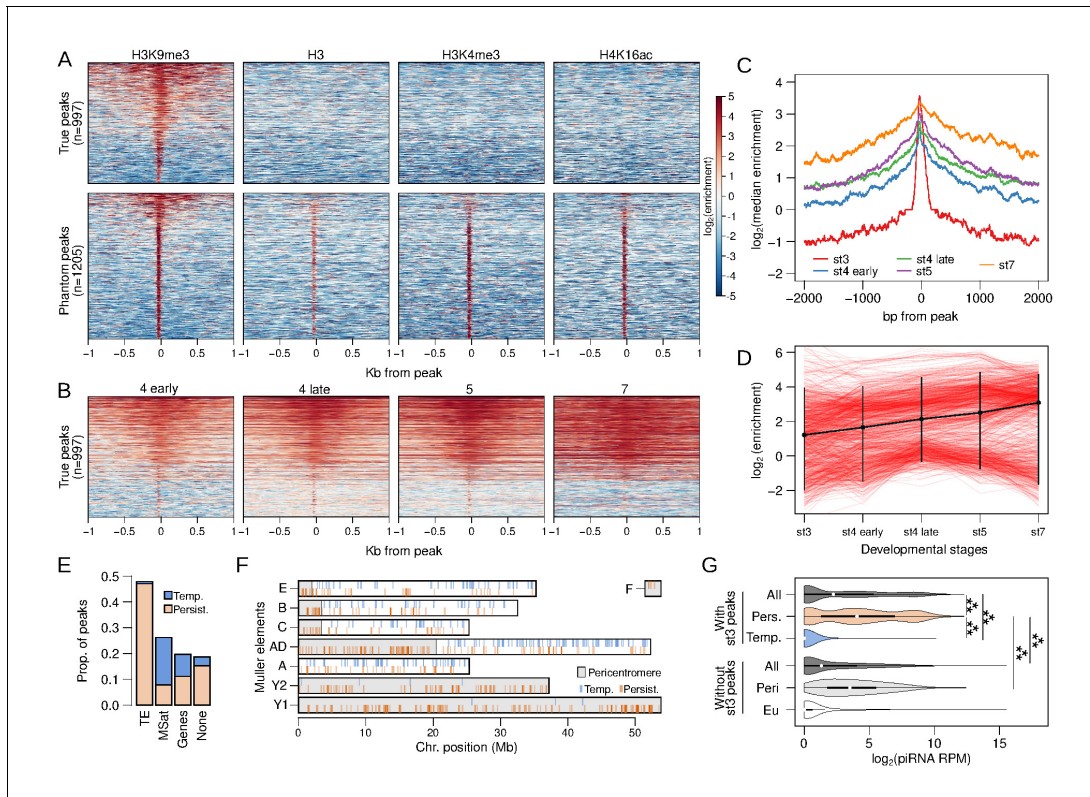

**Figure 3.** Stage 3 H3K9me3 peaks nucleate heterochromatin. (A) Separation of genuine stage 3 peaks from non-specific (phantom) stage 3 peaks using ChIP-seq against alternative histone modifications. Heatmaps depict the extent of enrichment around stage 3 H3K9me3 peaks (±1 kb). Peaks are sorted by the average H3K9me3 enrichment around each peak; therefore, all the different ChIP-seq enrichment plots have the same ordering. Top panels are H3K9me3-specific peaks, while bottom panels are non-specific peaks. (B) H3K9me3 enrichment around stage 3 peaks across developmental stages; peaks are ordered as in (A) (top). (C) Median H3K9me3 enrichment across developmental stages around stage 3 peaks. (D) Developmental trajectory of each H3K9me3 peak (red lines). Enrichment of a peak is estimated as enrichment averaged across ±100 bp around each peak. The average across all peaks is in black with error bars representing 95% confidence intervals. (E) Distribution of stage 3 peaks across different annotation categories. (F) Placement of peaks across the genome; gray regions mark the pericentric and heterochromatic regions of the genome. Persistent and temporary peaks are plotted on the bottom and top halves of each chromosome arm, respectively. (G) Distribution of piRNA mapping in 5 kb genomic windows that overlap stage 3 peaks (top) and windows that do not overlap stage 3 peaks (bottom). The two sets of windows are further subdivided into those with persistent (Pers.) and temporary (Temp.) stage 3 peaks and those in (Peri) and outside (Eu) the pericentromeric regions, respectively. ***$p<2.2e-16$ Wilcoxon rank sum test.

The online version of this article includes the following figure supplement(s) for figure 3:

**Figure supplement 1.** Examples of stage 3 peaks.

**Figure supplement 2.** RNA-seq reads from developmentally staged male embryos around stage 3 peaks.

**Figure supplement 3.** Transcript abundance of genes with stage 3 peaks.

**Figure supplement 4.** piRNA mapping around stage 3 peaks.

they are found scattered across the genome, as opposed to being concentrated near the pericentric regions (*Figure 3F*). Interestingly, 25.6% (n = 256) of the stage 3 peaks appear to become depleted of H3K9me3 enrichment over time (*Figure 3B,D*), possibly indicating that some nucleating sites fail to initiate and/or maintain heterochromatin spreading, but they could also be phantom peaks that remained after filtering. The bulk of these temporary peaks overlap microsatellites (71.4%) and only 3.1% overlap TEs (*Figure 3E*). In contrast, peaks that persist and expand through development (n = 741) mostly overlap TEs (63.3%), and only 10.5% are found at microsatellites (*Figure 3E*). Nearly all the temporary peaks are found outside of the pericentromere or the heterochromatic Y (*Figure 3F*).

H3K9me3 nucleation may be driven by RNA-mediated targeting (*Gu and Elgin, 2013*). We utilized embryonic RNA-seq data (*Lott et al., 2014*) and sequenced the piRNAs in 0–1 hr embryos to test for an association between H3K9me3 targeting and silencing of maternal or nascent TE transcripts by small RNAs. We find sparse RNA-seq reads from stage 2 or stage 4 embryos mapping to or around H3K9me3 peaks at embryonic stage 3 or 4 (*Figure 3—figure supplement 2*). Most of the RNA-seq reads around stage 3 peaks result from maternal transcripts of genes (*Figure 3—figure supplement 3*), with 40% of the 175 genes overlapping stage 3 peaks producing maternal transcripts. This is similar to the proportion of maternal genes genome-wide (39.8%; *Figure 3—figure supplement 3B*), arguing against the possibility that maternal transcripts induce nucleation. However, we find that genomic regions containing persistent stage 3 peaks have abundant piRNAs mapping around them (*Figure 3G*). Median piRNA reads per million (RPM) for 5 kb windows overlapping persistent stage 3 peaks is 15.55 (*Figure 3G*, *Figure 3—figure supplement 4*); this is substantially and significantly higher than for the rest of the genome (median RPM = 1.48), and also for pericentromeric windows lacking stage 3 peaks (medium RPM = 10.36; *Figure 3G*; Wilcoxon rank sum test p<2.2e-16). These results suggest that maternally deposited piRNAs play an important role in early H3K9me3 nucleation and are consistent with the model of piRNA-mediated co-transcriptional silencing (see below).

Temporary stage 3 peaks have almost no piRNAs mapping around them (median RPM = 0, *Figure 3G*). While this may simply reflect that temporary peaks are residual phantom peaks, it could also suggest that nucleation may involve other mechanisms independent or in addition to PIWI/piRNA-mediated targeting.

## Nucleation and spreading at early stage 4 are primarily at TEs and pericentric regions

While only about ~1000 H3K9me3 peaks are identified at stage 3, this number drastically increases to 52,607 in the early stage 4 samples (*Figure 4A*). Early stage 4 peaks show a gradual increase in levels of H3K9me3 enrichment and a local expansion of the H3K9me3 domain across development (*Figure 4A*, *Figure 4—figure supplement 1*). Unlike stage 3 peaks, nearly all early stage 4 peaks (97.7%) remain enriched throughout development. To better characterize this rapid burst of nucleation activity, we divided the early stage 4 peaks into those that already show enrichment at stage3 and those that do not, resulting in 16,630 and 35,977 peaks, respectively (*Figure 4—figure supplement 2*). The former (hence forth, old peaks) contain sites that are in the process of nucleation at stage 3, but their H3K9me3 enrichment is below the threshold used for peak-calling (*Figure 4—figure supplement 2A*). The latter (hence forth, new peaks) contain sites that began nucleating at stage 4, as they show no prior heterochromatin enrichment (*Figure 4—figure supplement 2B*).

Consistent with subsequent spreading of H3K9me3 after nucleation, old peaks are on average 847 bp, significantly wider than young peaks which are on average 378 bp (Wilcoxon rank sum test, two-tailed p-value<2.2e-16; *Figure 4B*). Furthermore, secondary peaks can be observed around the old peaks, indicating the deposition of H3K9me3 to adjacent nucleosomes by histone methyltransferases (*Figure 4C*). In contrast, the new peaks show a single summit flanked by depletion, consistent with H3K9me3 at single nucleosomes. Both sets of peaks increase in H3K9me3 enrichment across developmental stages with the new peaks having significantly lower enrichment across all stages; however, the difference in enrichment between the two sets of peaks decreases over time (*Figure 4D*). The gradual increase in H3K9me3 enrichment across stages likely reflects increasing numbers of nuclei/cells in the embryo with H3K9me3 deposited around nucleating sites as heterochromatin becomes stably established, with older peaks reaching saturation as the new peaks catch up. To determine whether increases in H3K9me3 enrichment are associated with expected decreases

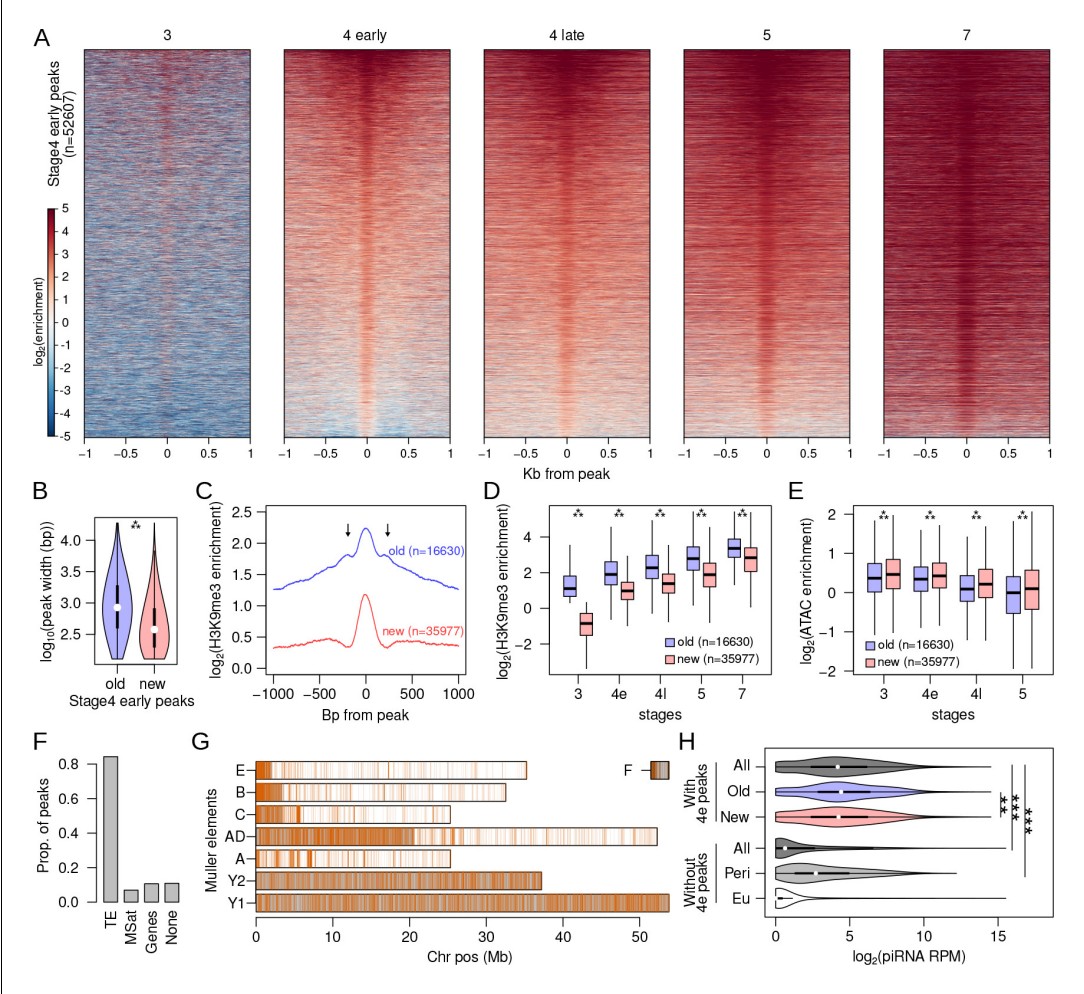

**Figure 4.** Rapid nucleation in early stage 4. (**A**) H3K9me3 enrichment around (±1 kb) early stage 4 peaks across development. Peaks are sorted by mean enrichment at early stage 4. (**B**) Peak widths of early stage 4 peaks that show enrichment in stage 3 (old) and peaks that show no enrichment in stage 3 (new); ***p<2.2e-16 (Wilcoxon rank sum test). (**C**) Median H3K9me3 enrichment around old and new early stage 4 peaks. Arrows mark secondary peaks around the old peaks. (**D**) H3K9me3 enrichment of old and new early stage 4 peaks across development; ***=p < 2.2e-16 (Wilcoxon rank sum test). (**E**) Accessibility as measured by ATAC-seq enrichment in old and new peaks; **p<2.2e-16 (Wilcoxon rank sum test). (**F**) Distribution of new early stage 4 peaks across different annotation categories. (**G**) Genome-wide distribution of new early stage 4 peaks; gray regions mark the pericentric and heterochromatic regions of the genome. (**H**) Distribution of piRNA mapping in 5 kb genomic windows that overlap (top) and do not overlap (bottom) early stage 4 peaks. The windows are further subdivided into containing old or new peaks and being located in (Peri) and outside (Eu) the pericentromeric regions, respectively. **p<0.0001, ***p<2.2e-16 Wilcoxon rank sum test.

The online version of this article includes the following figure supplement(s) for figure 4:

**Figure supplement 1.** Examples of stage 4 early old (A, B) and new (C, D) peaks.

**Figure supplement 2.** H3K9me3 enrichment at early stage 4 old and new peaks across development.

**Figure supplement 3.** RNA-seq reads from developmentally staged male embryos around stage 4 new peaks.

**Figure supplement 4.** piRNA mapping around early stage 4 peaks.

in chromatin accessibility, we generated ATAC-seq libraries from single embryos for the same developmental stages (absent stage 7). Indeed, we find that both sets of peaks show decreasing accessibility over development, with the old peaks being significantly less accessible across all stages (*Figure 4E*).

Expectedly, most of the stage 4 peaks (84.2%) overlap with annotated TEs (*Figure 4F*) and 82.6% of them are concentrated at the pericentromere and the repeat-rich Y chromosome (*Figure 4G*). These distributions are drastically different from stage 3 peaks, revealing that nucleation at stage 4 is more localized to canonically heterochromatic sites and TEs. Again, we looked for associations

of stage 4 peaks with transcripts and piRNAs. As with the stage 3 peaks, very few stage 2 or 4 embryonic RNA-seq reads map to or around the peaks (*Figure 4—figure supplement 3*). Similar to persistent stage 3 peaks, we find that genomic (5 kb) windows with stage 4 peaks have abundant piRNAs mapping, with median RPM of 17.21 (*Figure 4—figure supplement 4*). Again, this is significantly higher than both windows without peaks (median RPM = 0.56, Wilcoxon's rank sum tests, p<2.2e-16) or windows in the pericentromere but without peaks (median RPM = 5.55, Wilcoxon's rank sum tests, p<2.2e-16; *Figure 4H*). Furthermore, genomic windows with old stage 4 peaks have slightly but significantly more piRNA (median RPM = 20.36) than new peaks (median RPM = 17.95; Wilcoxon's rank sum tests, p=6.57e-05; *Figure 4H*). These results again show that heterochromatin establishment during early embryogenesis is associated with maternal piRNAs and support a model of PIWI/piRNA-mediated targeting of heterochromatin. However, many windows with stage 4 peaks have no piRNAs mapping, and many windows without peaks have high piRNAs mapping (*Figure 4H*), suggesting that additional signals are likely at play (see below).

## Heterochromatin nucleation at TEs is restricted at stage 3 but wide at stage 4

As TEs dominate stage 3 and early stage 4 nucleation sites (*Figures 3E* and *4F*), we further characterized patterns of nucleation and spreading at specific TE families annotated across the genome. Of the 235 entries in the curated TE library of the *D. pseudoobscura* group (*Hill and Betancourt, 2018*), only 35 (14.8%) overlap with stage 3 peaks while 189 (80.3%) overlap with the early stage 4 peaks (*Figure 5A*). Stage 3 peaks reside predominantly in retrotransposons (*Figure 5A*), and can be found both within non-LTR retrotransposons (e.g. LOA and R1) and LTR retrotransposons (e.g. Bel, Gypsy11, TRAM). In contrast, stage 4 peaks show a very different distribution among TEs (*Figure 5A*). For example, three TE variants (CR1-1, LOA-3, and Gypsy18) have the most early stage 4 peaks but few stage 3 peaks (*Figure 5A*), suggesting that some TEs nucleate heterochromatin earlier than others.

H3K9me3 nucleation within TEs could occur at specific positions or be distributed more evenly across the TE. We looked at the placement of H3K9me3 peaks with respect to the full-length TE consensus sequence and overall H3K9me3 enrichment of all the annotated insertions for each TE in the *D. miranda* genome. For TEs with large numbers of stage 3 peaks, the peaks are typically highly restricted to a specific region of TEs. For example, the vast majority of stage 3 peaks localize between 3500 and 4000 bp of the R1-6 element (*Figure 5B*). This starkly contrasts from the early stage 4 peaks, which are scattered across the entirety of the R1-6 element (*Figure 5C*). As expected, the position of the cluster of stage 3 peaks also corresponds to the region with the highest H3K9me3 enrichment across the R1-6 element, producing spikey and heterogeneous H3K9me3 enrichment profiles at stage 3 (*Figure 5B–D*). However, at later stages, H3K9me3 enrichment becomes more evenly elevated around the peak region, indicative of spreading of heterochromatin for robust silencing during development (*Figure 5D*). Similarly, the TRAM element shows highly restricted H3K9me3 enrichment and peaks localized to the 5' (0–300 bp) region at stage 3 followed by broad heterochromatin enrichment at early stage 4 and peaks spread throughout the element (*Figure 5E–G*). The 5' region of TRAM, where the early nucleating peaks reside, continues to increase in H3K9me3 over development as enrichment levels at the rest of the TE (*Figure 5G*). Notably, the 5' region of nearly all of the 748 TRAM copies found in the *D. miranda* genome show elevated H3K9me3 enrichment even though only 27 peaks have been called there (*Figure 5E*), indicating that peak calling is highly conservative and substantially underestimates the number of nucleation sites. This pattern of localized stage 3 nucleation is also observed in other TEs (*Figure 5—figure supplement 1*) and for LTR-retrotransposons, nucleation appears restricted to the 5' end of the TE (*Figure 5—figure supplement 2*).

For TEs like CR1-1 that have few to no stage 3 peaks, H3K9me3 establishment at early stage 4 is nonetheless very broad (*Figure 5H–J*, *Figure 5—figure supplement 1*). When looking at H3K9me3 enrichment at stage 3 across all CR1-1 insertions, we noticed low but consistent patterns of localized H3K9me3 enrichment at multiple positions of the TE (e.g. ~1800 bp, ~2200 bp, and ~4000 bp); these regions also show some of the most elevated H3K9me3 enrichment at later stages (*Figure 5J*, *Figure 5—figure supplement 1*). These patterns suggest that despite the absence of H3K9me3 peak calls at stage 3, localized nucleation likely is in progress at this stage, but possibly in only a subset of nuclei, causing low overall enrichment. Thus, we conclude that nucleation begins first as targeted

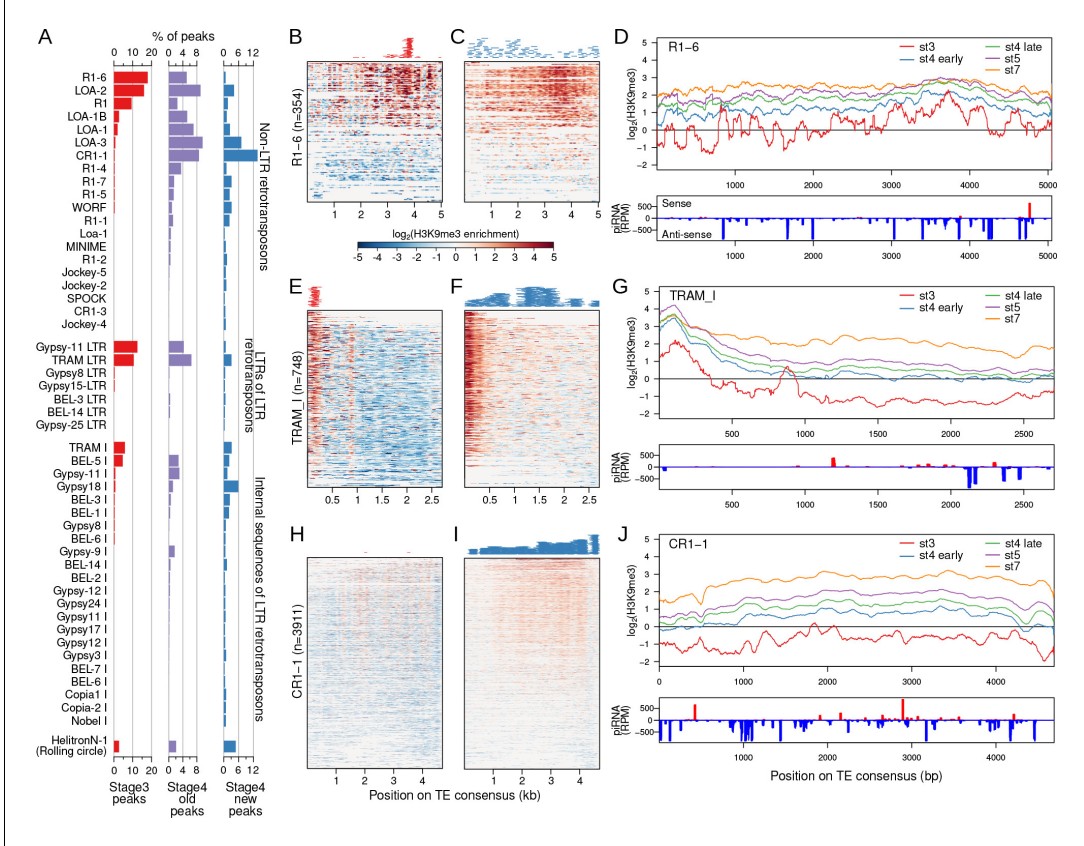

**Figure 5.** Narrow nucleation followed by wide establishment of heterochromatin at TEs. (**A**) TEs with annotated insertions overlapping stage 3, stage 4 old peaks, and stage 4 new peaks are listed; barplots depict the proportion of peaks overlapping with each TE. (**B**) H3K9me3 enrichment at stage 3 at all annotated TE insertions of the TE R1 variant (R1-6). Full length and fragmented annotated insertions are lined up with respect to their positions on the consensus TE sequences. Insertions are sorted by average enrichment. Positions of the called peaks are plotted above the heatmap. (**C**) Same as (**B**), but for stage 4 enrichment and new peaks. (**D**) Top. Mean enrichment of all insertions for the TE across development. Bottom. Sense and anti-sense piRNA mapping across the TE. (**E–G**) and (**H–J**), same as (**B**, **C**) but for the TEs TRAM and CR1-1, respectively. For more examples of enrichment over TEs, see *Figure 5—figure supplements 1* and *2*.

The online version of this article includes the following figure supplement(s) for figure 5:

**Figure supplement 1.** Examples of non-LTR TE families with abundant nucleation at stage 3 (**A, B**) and/or stage 4 (**C, D**).

**Figure supplement 2.** Examples of LTR retrotransposons with abundant nucleation in either the LTR sequence or the internal sequence.

H3K9me3 enrichment at TE insertions followed by wide H3K9me3 deposition and spreading across the rest of the element.

## High maternal piRNA abundance and early zygotic transcription are associated with early nucleation at TEs

The piRNA pathway allows for sequence-specific targeting of TEs for both co-transcriptional silencing and post-transcriptional degradation (*Malone and Hannon, 2009*). Indeed, we observed that nucleating peaks tend to be found in genomic regions with high piRNA mapping (*Figures 3G* and *4H*). To further evaluate whether piRNAs play a role in the targeted nucleation at TEs, we looked at the amount of maternally deposited sense and anti-sense piRNA mapping specifically to TEs (instead of genomic regions around peaks). Interestingly, we find that TEs with stage 3 peaks generate more maternal sense and anti-sense piRNAs than other TEs (*Figure 6A*; p=4.2e-06 and p=0.000118, respectively, Wilcoxon rank sum test). TEs with stage 4 peaks are also enriched for piRNAs, but to a lesser extent than stage 3 peaks (*Figure 6A*). Therefore, TEs generating abundant piRNAs - also tend to be the earliest to be targeted for H3K9me3 nucleation. piRNA abundance is much more strongly and significantly correlated with H3K9me3 enrichment of TEs at the later developmental

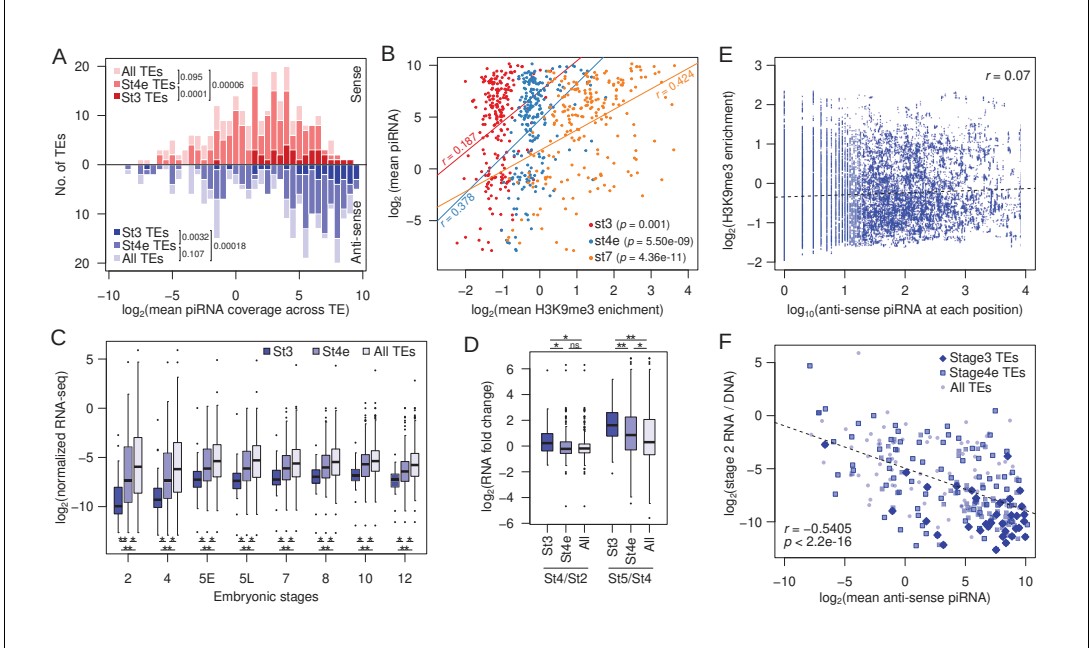

**Figure 6.** Association between early H3K9me3 nucleation at TEs, maternal piRNA production, and expression. (**A**) Distribution of TEs by their average maternally deposited sense (top, red) and anti-sense (bottom, blue) piRNA coverage. TEs with stage 3 peaks (stage 3 TEs), early stage 4 peaks (stage 4e TEs), and all TEs are in decreasing color intensity. p-values of pairwise comparisons determined by Wilcoxon's rank sum test are marked beside the legends. (**B**) Correlation between average H3K9me3 enrichment across a TE at different developmental stages and average piRNA abundance. Pearson's correlation coefficients (r) are labeled beside the regression lines. (**C**) Average expression of TEs with stage 3 peaks, early stage 4 peaks, and all TEs across embryonic development. Pairwise significance is determined by Wilcoxon's rank sum rest: *p value<0.05 and **p value<0.001 after multiple-testing correction with false discovery rate. (**D**) Zygotic expression of different TE classes is approximated by the fold difference between early embryonic stages. Pairwise significance determined as in (**C**). (**E**) For each position of stage 3 nucleating TEs, the H3K9me3 enrichment is plotted against the number of piRNA reads mapped. Linear regression is plotted in dotted line and Pearson's correlation coefficient (r) is labeled. (**F**) Negative correlation between maternally deposited anti-sense piRNA and expression of all TEs (light blue dots), stage 3 TEs (dark blue diamonds), and early stage 4 TEs (blue boxes). Expression of TEs is scaled by copy number by dividing the RNA-seq read counts with DNA-seq read counts. Dotted line demarcates the linear regression and r represents the Pearsons' correlation coefficient.

The online version of this article includes the following figure supplement(s) for figure 6:

**Figure supplement 1.** Density scatter plot of the correlation between H3K9me3 enrichment of new stage 4 peaks against maternally deposited piRNA coverage.

stages (***Figure 6B***), which could indicate that piRNAs further facilitate the spreading and maturation of H3K9me3-dependent silencing after early nucleation.

Co-transcriptional silencing of repeats in *Drosophila* requires transcription of TEs (***Verdel et al., 2004***; ***Ozata et al., 2019***). We, therefore, examined TE expression in developmentally staged (stages 2–12) and sexed single embryo RNA-seq datasets (***Lott et al., 2014***); note that while these data measure standing levels of transcripts, increases and decreases in RNA transcript levels across stages reflect zygotic transcription, or RNA degradation, respectively. Across all developmental stages, transcript abundances of TEs with stage 3 peaks are significantly lower than the remaining TEs (***Figure 6C***). The vast majority of transcripts in stage 2 embryos are maternally deposited, and TE expression increases across the board as zygotic expression ramps up (most noticeable between stages 4 and 5, ***Figure 6C***). However, TEs with stage 3 nucleation sites show the highest and earliest increase in transcript abundance (***Figure 6C,D***). While transcript abundances of most TEs, on average, remain unchanged between stages 2 and 4 (0.97-fold difference), stage 3 nucleating TEs show a 1.41-fold increase in RNA abundance from stage 2 to stage 4 embryos (***Figure 6D***). Interestingly, TEs with stage 4 nucleation peaks, although showing no change in expression between stages 2 and 4 (1.01-fold difference), start showing evidence of zygotic transcription slightly later, when contrasting transcript abundances between zygotic stages 4 and 5 (***Figure 6C,D***). These results reveal that

early nucleating TEs are among the first to be zygotically transcribed during embryogenesis, consistent with the need for nascent TE transcripts to induce PIWI-associated co-transcriptional silencing (*Shimada et al., 2016*).

While piRNA abundance is strongly associated with early nucleation, regions with abundant piRNA mapping on TEs do not necessarily correspond to the positions of stage 3 peaks nor elevated H3K9me3 enrichment (*Figure 5D,G,J*). At any given position along TEs with stage 3 peaks, the amount of piRNA mapping shows little-to-no correlation with H3K9me3 enrichment (*Figure 6E*). Furthermore, the extent of H3K9me3 enrichment does not correlate with the amount of piRNA mapping around peaks (*Figure 6—figure supplement 1*). These results argue against a simple targeting model where piRNA/PIWI complexes induce chromatin changes at the exact same sequence as where they associate with nascent transcripts, and suggest that additional targeting mechanism(s) are likely directing restricted H3K9me3 deposition within early nucleating TEs.

## Loss of nucleating sites at 5' LTR diminishes heterochromatin establishment

Early nucleating LTR retrotransposons, such as TRAM, appear to have most of their peaks and elevated H3K9me3 enrichment in their 5' region (*Figure 5B*, *Figure 5—figure supplement 2*). This, along with early zygotic expression, raises the possibility that transcription initiation may be involved in nucleation. TRAM is flanked by two 372 bp long terminal repeats (*Steinemann and Steinemann, 1997*; *Figure 7—figure supplement 1*), which contain numerous stage 3 peaks (*Figure 5A*). For LTR retrotransposons, the 5' LTR contains the primary promoter (*Thompson et al., 2016*). Indeed, the 5' LTR of TRAM is highly enriched for H3K9me3 (*Figure 7A*, *Figure 7—figure supplement 1*) and, to a lesser extent, also the 3' LTR (*Figure 7A*, *Figure 7—figure supplement 1*); we suspect the lower 3' enrichment mostly reflects unavoidable non-unique mapping between identical sequences within the LTR. In addition, LTR retrotransposons can form head-to-tail tandems such that a 3' LTR is also the 5' LTR of a downstream element (*McGurk and Barbash, 2018*; Ke and Voytas 1997), a structure that we indeed find in the *D. miranda* genome (*Figure 7—figure supplement 1*). Other LTR retrotransposons show a similar enrichment of stage 3 peaks in their 5' LTR (i.e. Gypsy-11, *Figure 5—figure supplement 2*).

To determine whether the loss of the 5' LTR may perturb the establishment of heterochromatin, we identified TRAM insertions with 5' truncations (n = 49), abolishing the 5' LTR and nucleating positions, and compared their H3K9me3 enrichment across development to that of full-length insertions (n = 392) (*Figure 7A*). We find that 5' truncated copies are significantly less enriched for H3K9me3 at their homologous regions after stage 3 with the difference in enrichment between 5' truncated and full-length copies increasing throughout development (*Figure 7A,B*); at stage 7, full-length inserts are on average 1.36-fold more enriched than 5' truncated inserts. We note that the difference in enrichment is likely to be a severe underestimate, since non-unique cross-mapping between the different copies (which is inevitable for repeats, especially for a highly abundant recently active one; see below) curtails the difference in read coverage between copies; the difference in enrichment therefore relies on reads that gap insertion junctions which are unique for different insertions. Additionally, we expect heterochromatic spreading from neighboring repeats to further minimize the difference in enrichment.

Notably, diminished H3K9me3 enrichment is not observed from 3' truncated TRAM copies (n = 51, *Figure 7—figure supplement 1*) where the 5' LTR remains intact (*Figure 7C*, *Figure 7—figure supplement 2*); in fact, 3' truncated TRAM copies have elevated enrichment compared to full-length copies as they, by necessity, contain proportionally more of the highly enriched 5' sequences (*Figure 7—figure supplement 2*). Further supporting the functional importance of the promoter adjacent nucleating sites for heterochromatin establishment, we find that 5' truncated insertions of TRAM tend to be more accessible based on ATAC-seq enrichment; albeit, the differences are either marginally significant or insignificant (*Figure 7D*).

## Early nucleating TEs are robustly silenced but were recently active

The amount of piRNA mapping to TEs is significantly anti-correlated with maternally deposited transcript abundance, particularly when scaled by TE copy number (Pearson's correlation coefficient = −0.54, p<2.2e-16; *Figure 6F*). TEs with stage 3 nucleation are associated with high

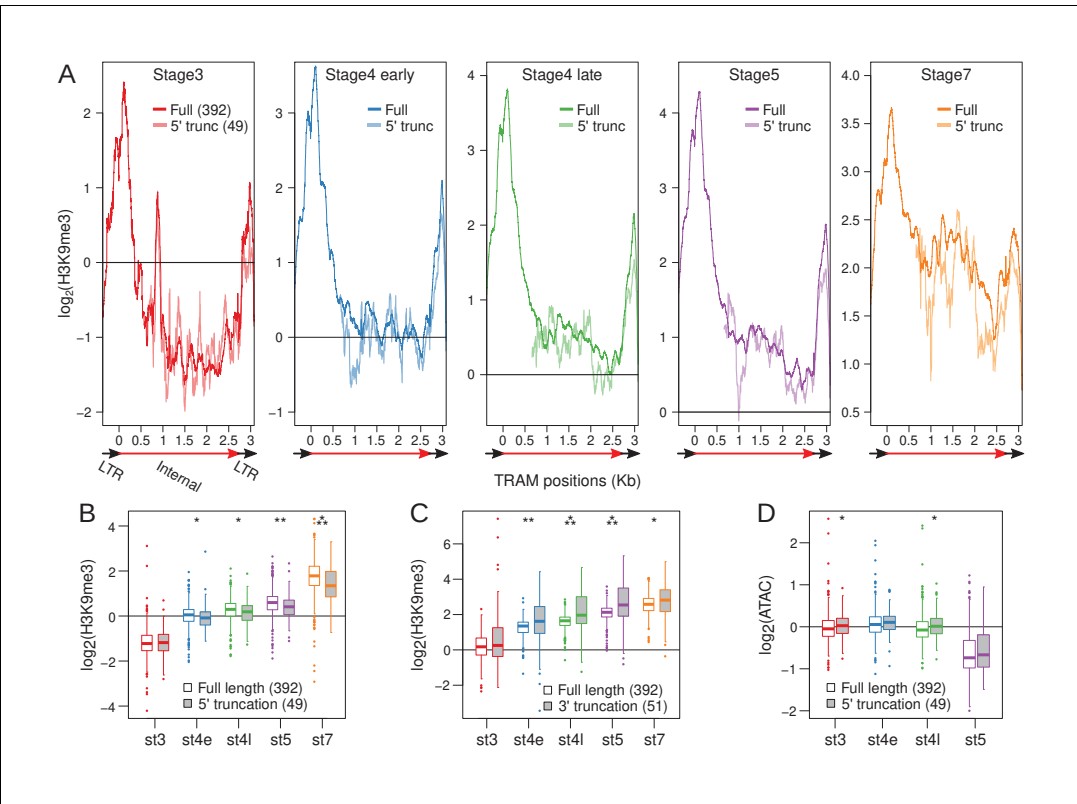

**Figure 7.** Loss of 5' LTR and nucleation sites reduce H3K9me3 enrichment at TRAM insertions. (A) H3K9me3 enrichment averaged across full-length TRAM insertions (sold lines) and 5' truncated TRAM insertions lacking the 5' nucleation sites (dotted lines) across developmental stages. Structure of TRAM is labeled below the plots (also see *Figure 7—figure supplement 1*). Note that the Y-axes change across the plots. (B) Distribution of average H3K9me3 enrichment for full length (boxes with white fill) and 5' truncated (boxes with gray fill) insertions across development. For each insertion, enrichment is averaged across the last 1 kb of the internal sequence (3' LTR is excluded). Boxplots depict the distribution of H3K9me3 enrichment across insertions. (C) Same as (B), but averaged across the first 1 kb of the internal sequence of full length and 3' truncated insertions (5' LTR is excluded). (D) Same as (B), but with ATAC enrichment instead of H3K9me3 enrichment. *p<0.05, **p<0.005, ***p<0.00005, Wilcoxon's rank sum test.

The online version of this article includes the following figure supplement(s) for figure 7:

**Figure supplement 1.** Structure and heterochromatin enrichment at TRAM.

**Figure supplement 2.** H3K9me3 enrichment in 3' TRAM truncations.

piRNAs counts (*Figure 6A*) and, as expected, significantly overrepresented with lowly expressing TEs (*Figure 6C*; p=1.582e-08, Wilcoxon's rank sum test); they remain to be lowly expressed as zygotic transcription increases after stage 4, indicating that early nucleating TEs are among the most repressed TEs (*Figure 6C*). TEs with stage 4 peaks show intermediate patterns with regards to piRNA counts and expression (*Figure 6A,C*).

High piRNA production and H3K9me3 enrichment and low transcript abundance of the early nucleating TEs suggest that they are targeted for robust silencing. These TEs may, therefore, have high transposition rates and impose a high fitness burden, creating strong selective pressure to evolve and strengthen epigenetic silencing mechanisms. Indeed, we find that the early nucleating TEs are significantly more abundant in both the male and female *D. miranda* genomes (*Figure 8A*). Because genome-wide TE abundance can reflect ancient accumulation and not just recent transposition rates, we also looked at insertions specifically on the neo-Y, which more than doubled in size due to the expansion of TEs since its formation 1.5MY ago (*Mahajan et al., 2018*). Consistent with recent activity, the neo-Y has on average 1.79-fold more early nucleating TEs than the rest of the genome (*Figure 8B*). These results reveal that early nucleating TEs have been recently active in the

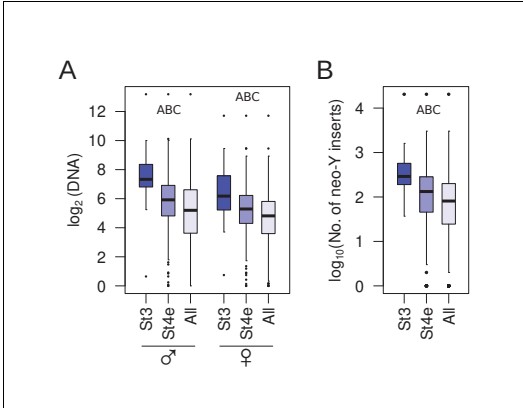

**Figure 8.** Genomic abundance of early nucleating TEs. (A) Comparisons between copy number abundance of stage 3 nucleating, early stage 4 nucleating, and all TEs in males and females. (B) Same as (A), but with number of insertions on the neo-Y. Pairwise significance is represented by letters where -upper-case letters denote p-value<0.001 (Wilcoxon rank sum test); A = stage 3 TEs vs. early stage 4 TEs, B = early stage 4 TEs vs. all TEs, and C = stage 3 TEs vs. all TEs.

*D. miranda* lineage, generating a large number of insertions throughout the genome. The strong genomic defense through early nucleation and high piRNA production may therefore have evolved to minimize their deleterious activity.

## Discussion

### H3K9me3-dependent heterochromatin nucleation in early development

In metazoans, extensive epigenetic reprogramming occurs during gametogenesis and early embryogenesis where most chromatin marks, including H3K9me3, are erased and re-established later (*Ishiuchi et al., 2015*; *Wang et al., 2018*; *Laue et al., 2019*). H3K9me3 re-establishment in somatic tissues is crucial for normal development in *Drosophila* and mammals. A number of different pathways have been identified that recruit H3K9 methyltransferases to target loci and include a diversity of guides such as DNA binding proteins and non-coding RNAs. For example, in fission yeast, small interfering RNAs (siRNAs) matching pericentric repeats are required for recruitment of the silencing machinery to pericentric heterochromatin (*Volpe et al., 2002*; *Hall et al., 2002*). In mammals, a parallel pathway exists based on DNA sequence-based recognition and silencing of mobile elements by KRAB- zinc-finger DNA-binding proteins. These proteins bind DNA via an array of zinc-fingers and recruit the universal co-repressor KAP1 (KRAB-associated protein 1) through their KRAB domain. In turn, KAP1 recruits histonemethyltransferases to TE insertions to promote deposition of repressive chromatin marks (*Wolf et al., 2015*).

The role of H3K9me3 in early somatic development has been studied in mouse, and H3K9me3-dependent heterochromatin was shown to undergo dramatic reprogramming during early embryonic development (*Wang et al., 2018*; *Fadloun et al., 2013*). Targeting of H3K9me3 during fly embryogenesis is poorly understood. We find that establishment of heterochromatin begins with limited H3K9me3 deposition at single nucleosomes at or before stage 3. These nucleating sites expand gradually by H3K9me3 deposition to neighboring nucleosomes in subsequent stages to form mature heterochromatin. Unexpectedly, we find that early nucleation sites may not be restricted to the pericentric regions and can be lost through development. Nevertheless, the majority of nucleating sites are localized to specific regions of TEs such that across different insertions, the same positions show elevated H3K9me3 enrichment. Interestingly, truncated insertions that lost these positions show significantly reduced H3K9me3 enrichment across the remainder of the TE and maintain elevated accessibility across development, revealing that these nucleating positions are important for subsequent local spreading of heterochromatin.

### Potential mechanisms for targeted nucleation in TEs

The restricted positioning of nucleation suggests a targeted mechanism for H3K9me3 deposition. Although one of the key functions of heterochromatin is to induce transcriptional silencing at TEs, a low level of nascent transcription has been implicated in the initial formation of heterochromatin in several organisms (*Allshire and Madhani, 2018*). Transcription of satellites during embryogenesis has been shown to be required for the formation of heterochromatin in mouse and *Drosophila* (*Probst et al., 2010*; *Casanova et al., 2013*; *Mills et al., 2019*), and satellite transcription and/or RNAs are necessary for centromere and heterochromatin functions in humans, mice, and flies (*Rošić et al., 2014*; *Johnson et al., 2017*; *Shirai et al., 2017*; *Velazquez Camacho et al., 2017*; *McNulty et al., 2017*). Chromatin-associated transcripts may recruit silencing factors to their

genomic location (*Holoch and Moazed, 2015*), and co-transcriptional silencing by PIWIi–piRNA complexes has led to a model where piRNAs provide sequence specificity for PIWI to target nascent TE transcripts and guide co-transcriptional heterochromatin formation at TEs (*Holoch and Moazed, 2015*). In *Drosophila*, the piRNA pathway is thought to be primarily active in gonadal tissue, yet mutations of key factors in the pathway, including Piwi, affect heterochromatin formation in somatic tissues (*Pal-Bhadra et al., 2004*; *Gu and Elgin, 2013*). Both piRNAs and PIWI protein are maternally loaded into the egg, consistent with maternal piRNA/PIWI complexes guiding initial heterochromatin establishment in the early embryo, which is later maintained independently of piRNAs (*Pal-Bhadra et al., 2004*; *Gu and Elgin, 2013*).

Supporting the requirement of nascent transcription for heterochromatin formation, we find that TEs showing early nucleation have significantly higher zygotic expression than the rest during early development (between stages 2 and 4 for TEs with stage 3 peaks, or stages 4 and 5 for TEs with stage 4 peaks). In addition, we find that early nucleating TEs are associated with high abundance of both sense and anti-sense maternally deposited piRNAs. Early zygotic transcription during embryogenesis and high maternal piRNA counts are consistent with this model of co-transcriptional silencing of TEs and highlight the importance of small RNAs in establishing heterochromatin in flies.

While piRNA may guide the recruitment of the co-transcriptional repression machinery, it cannot account for the restricted nucleation of heterochromatin on TE inserts. We find that TEs have specific regions that are targeted first for heterochromatinization, and the positions of abundant piRNA read mapping show little correspondence to the positions of H3K9me3 nucleation along TEs. Our observation that early nucleating TEs also show early zygotic transcription raises the possibility that the act of transcription at these TEs may provide specificity; indeed, extensive chromatin remodeling occurs at promoters during zygotic genome activation (*Schulz and Harrison, 2019*). Supporting this, the strong H3K9me3 enrichment (and large number of peaks) around the 5' LTR of TRAM and other LTR-retrotransposons is consistent with a model where histone methyltransferases are recruited to the promoter of the TE insertions perhaps by transcription initiation. Clustering of H3K9me3 to the 5' end of a subset of TEs was also observed in mice (*Walter et al., 2016*).

However, this does not appear to be a general rule across all early nucleating TEs, as non-LTR elements like R1-6 and LOA-2 have restricted nucleation sites close to their 3' end or near the middle of insertions, respectively. These discrepancies may suggest that alternative targeting mechanisms are involved in early heterochromatin deposition in flies. KRAB- zinc-fingers are confined to mammals, but an analogous protein family exists in flies: ZAD- zinc-fingers (ZAD-ZNFs). ZAD-ZNFs are a poorly characterized gene family that has dramatically expanded within insect lineages (Chung et al. 2007; *Kasinathan et al., 2020*). Approximately half of all ZAD-ZNF genes in *D. melanogaster* are highly expressed in ovaries and early embryos (*Chung et al., 2002*) and several interact with heterochromatin (*Kasinathan et al., 2020*; *Swenson et al., 2016*). The ZAD-ZNF gene repertoire evolves rapidly across *Drosophila* species, which has been suggested to be driven by rapid alterations in heterochromatin across flies (*Kasinathan et al., 2020*). While it remains unclear how these proteins localize to heterochromatin, it raises the possibility that maternally deposited DNA-binding factors can recruit histonemethyltransferases for H3K9me3 deposition in flies. The localized nature of the early nucleating sites is suggestive of a targeting mechanism such as motif recognition by DNA-binding proteins. Future research in *Drosophila* will show whether specific sequences on TEs are targeted by maternally deposited DNA-binding proteins that act in concert with the piRNA pathway to cause site-specific H3K9me3 deposition at recently active TEs.

## Early and robust silencing against highly active TEs

Early nucleating TEs are under robust silencing; they appear sufficiently silenced during oogenesis with the lowest maternal transcript abundance, and their expression remains significantly lower than the remaining TEs as zygotic transcription ramps up. They are also disproportionately associated with high piRNA abundance and early zygotic transcription. What, then, sets these TEs apart and why are they more robustly silenced compared to others? Interestingly, we find that these TE are significantly more abundant with higher copy numbers and insertions across the genome. Elevated copy number on the neo-Y chromosome further indicates that these TEs had high transposition activity within the past 1.5 million years since the formation of the neo-Y. Altogether, these results suggest that the early nucleating TEs have high transposition potential and strong genomic silencing in the form of piRNA defense and targeted nucleation may have evolved to maintain genome

integrity. Indeed, many components of the piRNA machinery and heterochromatin/TE regulating proteins are rapidly evolving (*Blumenstiel et al., 2016*).

In the ongoing arms race between proliferation of selfish repeats and genome defense, early expression during zygotic genome activation can be beneficial for TEs to exploit immature heterochromatin (*Wei et al., 2020*). However, targeted early H3K9me3 nucleation may provide a strategy of counter-defense that complements piRNA-mediated transcriptional and post-transcriptional silencing.

## Materials and methods

### Embryo collection and staging

*D. miranda* adults were allowed to lay for 1 hr in an embryo collection cage at 18°C to pre-clear any old embryos that females may be holding. After pre-clearing, the embryos were aged to their appropriate developmental stage. Embryos were then dechorionated with 50% bleach solution for 1 min and identified according to its morphological features under a light microscope. Embryo developmental time/stage: stage 3 (210 min, polar buds at the posterior pole of the embryo), early stage 4 (240 min, thin 'halo' appears as the syncytial blastoderm forms), late stage 4 (270 min, thicker 'halo' compared with early stage 4 and no formation of cell membranes), stage 5 (300 min, visible cell membranes as cellularization progresses), stage 7 (420 min, cephalic furrow).

### Single embryo chromatin immunoprecipitation and library preparation with spike-in

The H3K9me3 ULI-nChIP preparation and pull down were done according to a modified version of *Brind'Amour et al., 2015*. In short, *D. miranda* embryos single embryos were homogenized into a cell suspension using a 200 µl pipette tip in 20 µl of Nuclei EZ lysis buffer (Sigma-Aldrich, Cat. No. N3408). To fragment the chromatin, 2 U/µl of MNase to were added into each nuclei preparations for 7.5 min at 21°C; reaction was terminated by adding 11 µl of 0.1M EDTA. Simultaneously, *D. melanogaster* stage 7 embryos were prepared the same way, and 5 µl was added into each *D. miranda* samples for spike-in. Then the digested nuclei preps were diluted to 400 µl with ChIP buffer (20 mM Tris–HCl pH 8.0, 2 mM EDTA, 150 mM NaCl, 0.1% Triton X-100, 1× Protease inhibitor cocktail) and split into a 40 µl aliquat and 360 µl aliquat for input and ChIP, respectively. Prior to the ChIP pull down, antibody against H3K9me3, (Diagenode, Cat No. C15410056), H4 (Abcam ab1791), H3K4me1 (Diagenode, Cat. No. C15410194), H4K16ac (Millipore, Cat. No. 07–329) were incubated with Dynabeads (ThermoFisher, Cat No. 10002D) for 3 hr to make the antibody-bead complex, followed by overnight incubation with the chromatin preparations. DNA from ChIP and input samples were isolated with Phenol-chloroform using MaXtract High Density tubes (Qiagen, Cat. No. 129046) to maximize sample retrieval, followed by ethanol precipitation and resuspension in 30 µl of nuclease-free water. The resulting DNA extractions were further purified to remove excess salt using Ampure XP beads at 1.8:1 beads-to-sample ratio (54 µl) and eluted in 10 µl of nuclease-free water. Library preparations were done using the SMARTer Thruplex DNA-seq kit from Takara (cat no. R400674). Library concentration was determined using Qubit (Thermofisher), and the library quality was determined using the Bioanalyzer by the Functional Genomics Lab at UC Berkeley.

### ChIP sequencing and data processing

All libraries were (100 bp) paired-end sequenced with the Hi-Seq 4000 at the Vincent J. Coates Genomics Sequencing Lab at UC Berkeley. To differentiate between spike-in and sample from each ChIP-seq library, raw paired-end reads are aligned to a reference file that contains both the *D. miranda* genome (r2.1) and the *D. melanogaster* genome (r6.12) using bwa mem (*Li and Durbin, 2009*) on default settings. Since bwa mem, by default, only reports the highest quality alignment, each read will align to one of the two genomes only once. When there are more than one alignment of equal mapping quality, the read will be randomly assigned to one of the targets. Using a custom Perl script, reads are then differentiated based on which of the two genomes they align to, and ambiguous alignments are discarded. For read mapping statistics, see *Supplementary file 1*. Reads are then sorted using samtools (*Li et al., 2009*; *Li and Durbin, 2009*), and the coverage per site is determined with bedtools genomecoverge (*Quinlan and Hall, 2010*) (-d -ibam options).

Sex of embryos were inferred from the ratio of the median autosomal coverage: median × coverage of the inputs (*Supplementary file 1*). Median coverages were inferred from distribution of average coverages in 1 kb sliding windows.

## Sample and enrichment normalization

For normalization without spike-in, we first took the average coverage of the ChIP and input samples in 1 kb windows and then determined the median of all the autosomal windows to obtain the median autosomal coverage. The per-site coverages in the ChIP and input samples were then divided by their respective median autosomal coverage, thus normalizing for library size. The per-site enrichment is then estimated as:

Enrichment = (ChIP/ChIP median + 0.01) / (input/input median + 0.01), with 0.01 being a small pseudo-count.

For normalization with spike-in, we first generated a reference spike-in enrichment profile by using the same procedure as above for all spike-in ChIPs and spike-in inputs in 1 kb windows, and then took the average $\log_2$ enrichment across all spike-ins, generating a reference. This reference is meant to be a standard to which all other spike-ins will be normalized, and the extent of normalization in the spike-ins will be applied to the actual samples. To do so, we used a method akin to quantile normalization, whereby the distribution of the spike-in sample is matched with that of the reference. However, instead of matching the distributions identically as with typical quantile normalization, we binned the enrichment values into quantiles of 0.1, allowing us to match the quantiles by bins. For example, if the 98.1th quantile has a $\log_2$ enrichment of 3.2 and 4.0 in the spike-in sample and reference, respectively, all points of the spike-in within the quantile will be increased by 0.8. Since the spike-in and the actual sample should have the same pull-down efficiency, this then allows us to determine the extent of normalization to apply to each $\log_2$ enrichment in the actual latter; for example, points with $\log_2$ enrichment of 3.2 will similarly be elevated by 0.8 in the sample.

## ChIP-seq peak calling

We used macs2 (*Zhang et al., 2008*) to call H3K9me3 enrichment peaks. For each stage containing multiple replicates, we called peaks using sorted bam files with the spike-in reads removed, where the spike-in reads are removed with the command below: macs2 callpeak -t replicate1.chip.sort.bam replicate2.chip.sort.bam... `-c replicate1.input.sort.bam replicate2.input.sort.bam... -n output_name -g 1.8e8 --call-summits`.

For peak calling in individual replicates, only the chip and corresponding input bam files are used for the -t and `-c parameters`. Peaks in different replicates that are less than 100 bp away from each other are deemed the same.

## Enrichment analysis around peaks

For each developmental stage, we averaged the H3K9me3 enrichment per site (or bin) across the replicates to generate a representative enrichment. To determine the developmental progression of enrichment around peaks, we averaged the representative enrichment at 100 bp upstream and downstream of the summit of the peak (reported by macs2) for each stage. Regions are deemed enriched if $\log_2$(enrichment) > 0.5. Enrichment on the neo-Y is averaged from only male embryos. For the enrichment heatmaps, we took the per-base enrichment 1000 bp upstream and downstream of the summits and sorted the peaks from highest enrichment to lowest enrichment. We then plotted each base as a dot using colors that scale with the enrichment with a custom R script. Colors are generated using the R package RColorBrewer. Heatmaps around peaks of ATAC-seq, RNA-seq, and DNA-seq are also generating using this method.

## Repeat annotations, contents, and overlap with peaks

We used Repeatmasker (*Smith et al., 2021*) with a TE index specific to the *obscura* group (the *Drosophila* lineage to which *D. miranda* belongs) from *Hill and Betancourt, 2018*. Simple repeats were identified using the simple repeat setting in Repeatmasker, which lists them and can be identified in the annotation file (.gff) as (motif)n where the 'motif' is the repeating unit (e.g. (AGAT)n). To determine the %repeat content, we used the repeat masker annotation and determined the number of bases annotated by either a TE or a simple repeat in nonoverlapping sliding windows. To determine

overlap between annotated repeats and H3K9me3 peaks, we used bedtools intersect -a annotate-drepeats.gff -b macs2.peaks -wa -wb. Distance between repeats and peaks is inferred using the R package IRange (*Lawrence et al., 2013*).

## H3K9me3 enrichment at full length and truncated annotated TE insertions

The Repeatmasker annotation (gff) provides the positions of the TE insertions in the genome as well as the positions matching the TE consensus sequence; for each TE entry, these coordinates are used to create a table of enrichment values where each row is an annotated insertion in the genome and each column is a position of the consensus from the repeat index. Because the LTR and internal sequences of LTR retrotransposons are separate entries in the repeat index, we identified full-length insertions of TRAM in the genome by identifying internal sequences flanked by LTRs requiring that all three features must have the same direction (all forward/reverse) and are within 10 bp of each other. To identify 5′ and 3′ truncated insertions, we identified internal sequences where the respective LTRs are either missing or greater than 5 kb away. For statistical comparisons, we averaged the H3K9me3 enrichment across the last 1 kb of the element of each full length and truncated insert and compared the distribution of averaged enrichment between full length and 5′ insertions. The opposite was done for 3′ truncated insertions, where we averaged across the first 1 kb.

## DNA-seq and RNA-seq data analysis

Sexed DNA-seq and RNA-seq data are from *Mahajan et al., 2018* and *Lott et al., 2014*. Raw reads were aligned using bwa mem on default settings to either the *D. miranda* genome or the Repeat index (*Hill and Betancourt, 2018*) and sorted with samtools. After samtools sort, the per-base coverage is determined with bedtools genomeCoverageBed and then normalized by the number of mapped reads in the library. For copy-number-scaled TE transcript abundance, we tallied the RNA-seq and DNA-seq read count mapping to each TE entry in the repeat library; the former was normalized by the median read count at autosomal genes and the latter normalized by the median autosomal coverage. Mapping statistics for the RNA-seq samples can be found in *supplementary file 2*. For TE expression scaled by copy number, RNA-seq read counts were then divided by the DNA-seq read counts for each TE.

## Embryo collection for piRNA sequencing and piRNA isolation

Flies were allowed to lay on molasses plates with yeast paste for 1 hr, and embryos were immediately flash frozen in liquid nitrogen and stored at −80˚C. We used Trizol (Invitrogen) and GlycoBlue (Invitrogen) to extract and isolate total RNA. piRNA is isolated using $NaIO_4$ reaction and beta elimination with a protocol modified from (Kirino and Mourelatos 2007; Ohara et al. 2007; Cao et al. 2009; Simon et al. 2011; Abe et al. 2014; Gebert et al. 2015). We started with 20 µg total RNA in 13.5 µl water and added 4 µl 5× borate buffer (148 mM borax, 148 mM boric acid, pH 8.6) and 2.5 µl freshly dissolved 200 mM sodium periodate. After 10 min of incubation at room temperature, the reaction is quenched with 2 µl gylcerol for an additional 10 min. We poked holes in the tops of the tubes with sterile needles, then vacuum dried the reaction for 1 hr. We then added 50 µl 1× borax buffer (30 mM borax, 40 mM boric acid, 50 mM NaOH, pH 9.5, and 200 mM sodium periodate) and incubated the samples for 90 min at 45˚C. For RNA precipitation, we added 1.33 µl GlycoBlue (Invitrogen) and 200 µl ice cold 100% EtOH and place the samples in −80˚C overnight. We centrifuged down the samples at 4˚C, maximum speed, for 15 min, removed the supernatant, and let the RNA pellet dry, and resuspended the RNA in 20 µl $H_2O$.

Twenty microliters of RNA prepared as described above was resolved on a 15% TBE-urea gel (Invitrogen). Custom 18-nt and 30-nt ladders were used to size select 19–29 nt long RNA, which was purified and used as input for library preparation using the Illumina TruSeq Small RNA Library Preparation Kit to ligate adapters, reverse transcribe and amplify libraries, and purify the cDNA constructs. The protocol included an additional size selection step after library preparation on a 6% TBE gel (Invitrogen). Fifty base-pair single-end sequencing of our samples was performed on an Illumina HiSeq 4000 at the Vincent J. Coates Genomic Sequencing Laboratory at UC Berkeley.

## piRNA-seq data analysis

Adapters were trimmed from the reads using trim_galore (http://www.bioinformatics.babraham.ac.uk/projects/trim_galore/). We mapped the reads to the *D. miranda* genome and the repeat index using the small RNA-specific aligner ShortStack (v3.8.5) (*Johnson et al., 2016*) with settings `-bow-tie_m` all `-ranmax` none `-nohp`. Using samtools view and awk, we then extracted reads that are 23–29 bp in length. For mapping to the repeat index, we further used samtools view to separate forward and reverse mapping reads with the –F 16 and –f 16, respectively. We then used bedtools genomeCoverage to obtain piRNA coverage at each position in the genome or repeat index. piRNA reads mapping statistics can be found in *Supplementary file 3*. To determine the amount of piRNAs mapping to 5 kb windows across the genome, we used bedtools coverage. Overlap (and lack of) between the resulting bed files and peaks are then determined using bedtools intersect. RPM is calculated as the number of reads mapping to window/position divided by million of 23–29 bp piRNA reads mapped.

## ATAC-seq sample preparation

Single-staged embryos were collected as above. ATAC-seq preparation was done according to a modified protocol of *Buenrostro et al., 2015*. Embryos were homogenized into a cell suspension using a 200 µl pipette tip in 20 µl of Nuclei EZ lysis buffer (Sigma-Aldrich, Catalog No. N3408). The supernatant was removed after centrifugation at 4℃. The nuclei were the resuspended in the transposition buffer (2× reaction buffer, Illumina Cat #FC-121–1030, Nextera Tn5 Transposase, Illumina Cat #FC-121–1030) and PCR amplified (KAPA HiFi HotStart ReadyMix, cat# KK2600) for 12 cycles. PCR products were cleaned up with AMPure XP beads (cat# A63881) at 1:1 concentration. Samples were sequenced with the Novaseq 6000 S1 with 100 bp pair-end reads at the Vincent J. Coates Genomic Sequencing Lab at UC Berkeley.

## ATAC-seq data analysis

Pair-end reads are adapter trimmed with trimgalore and aligned to the reference genome with bwa mem on default settings. After sorting with samtools, we used bedtools genomeCoverageBed with the -pc flag for fragment count instead of read count, generating the per base fragment coverage. The fragment coverage for each sample is then normalized by the median autosomal coverage (*Supplementary file 4*). The sex of the embryos is then determined based on the coverage on the sex chromosomes; i.e., males will have ~0.5× coverage on the X and Y chromosomes. Since our analyses focused on repetitive heterochromatin where read coverage deviates highly due to copy number differences, we controlled for copy number by dividing the ATAC-seq fragment counts by fragment counts of sex-matched DNA-seq samples, generating ATAC enrichment. This also simultaneously controls for the coverage differences of sex chromosomes between males and females.

## Data availability

All ChIP-seq and ATAC-seq data generated have been deposited on SRA under BioProject PRJNA601450. Intermediate files, including ChIP enrichment files and peak calls, are uploaded on Dryad. R and perl scripts for spike in normalization, generating enrichment around peaks, and enrichment heatmaps are available on KW's github page (https://github.com/weikevinhc/heterochromatin.git, swh:1:rev:8a5f9761033beeccda8d6ed600cc67e45400c6f2); *Wei, 2021*.

## Acknowledgements

We thank Dr. Lauren Gibilisco for technical guidance with piRNA library preparation and analyses and Dr. Sally Elgin for comments on manuscript draft. We also thank the four anonymous reviewers for their valuable suggestions and critiques. Publication made possible in part by support from the Berkeley Research Impact Initiative (BRII) sponsored by the UC Berkeley Library.

## Additional information

### Funding

| Funder | Grant reference number | Author |
|---|---|---|
| National Institutes of Health | R56AG057029 | Doris Bachtrog |
| National Institutes of Health | R01GM101255 | Doris Bachtrog |
| National Institutes of Health | R01GM076007 | Doris Bachtrog |

The funders had no role in study design, data collection and interpretation, or the decision to submit the work for publication.

### Author contributions

Kevin H-C Wei, Conceptualization, Data curation, Software, Formal analysis, Validation, Investigation, Visualization, Methodology, Writing - original draft, Writing - review and editing; Carolus Chan, Data curation, Validation, Investigation, Methodology; Doris Bachtrog, Conceptualization, Resources, Data curation, Supervision, Funding acquisition, Methodology, Writing - original draft, Project administration, Writing - review and editing

### Author ORCIDs

Kevin H-C Wei  https://orcid.org/0000-0002-1694-9582
Doris Bachtrog  https://orcid.org/0000-0001-9724-9467

### Decision letter and Author response

Decision letter https://doi.org/10.7554/eLife.55612.sa1
Author response https://doi.org/10.7554/eLife.55612.sa2

## Additional files

### Supplementary files

- Supplementary file 1. ChIP-seq sample information and mapping statistics.
- Supplementary file 2. Embryonic RNA-seq sample information and mapping statistics.
- Supplementary file 3. piRNA sequencing mapping statistics.
- Supplementary file 4. ATAC-seq sample information and mapping statistics.
- Transparent reporting form

### Data availability

All ChIP-seq and ATAC-seq data generated have been deposited on Genebank under BioProject PRJNA601450. Intermediate files, including ChIP enrichment files and peak calls, are uploaded on Dryad. R and perl scripts for spike in normalization, generating enrichment around peaks, and enrichment heatmaps are available on KW's github page (https://github.com/weikevinhc/heterochromatin.git (copy archived at https://archive.softwareheritage.org/swh:1:rev:8a5f9761033beeccda8d6ed600cc67e45400c6f2)).

The following datasets were generated:

| Author(s) | Year | Dataset title | Dataset URL | Database and Identifier |
|---|---|---|---|---|
| Wei KHC, Chan C, Bachtrog D | 2020 | Restricted nucleation and piRNA-mediated establishment of H3K9me3-dependent heterochromatin during embryogenesis in *Drosophila miranda* | https://www.ncbi.nlm.nih.gov/bioproject/PRJNA601450 | NCBI BioProject, PRJNA601450 |
| Wei KHC, Chan C, Bachtrog D | 2020 | Data from: Establishment of H3K9me3-dependent | https://doi.org/10.6078/D1TH7J | Dryad, 10.6078/D1TH7J |

heterochromatin during embryogenesis in *Drosophila miranda*

The following previously published dataset was used:

| Author(s) | Year | Dataset title | Dataset URL | Database and Identifier |
|---|---|---|---|---|
| Lott SE, Villalta JE, Zhou Q, Bachtrog D, Eisen MB | 2014 | Sex-specific embryonic gene expression in species with newly evolved sex chromosomes | https://www.ncbi.nlm.nih.gov/bioproject/PRJNA232085 | NCBI BioProject, PRJNA232085 |

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
