## [Decision Letter]

**Acceptance summary:**

In this study, the authors characterize H3K9me3 establishment during embryonic development in *D. miranda*. They identify a subset of H3K9me3 peaks, frequently proximal to transposon-targeted by piRNAs, that progressively broaden into stable heterochromatin at Stage 7 of development. Thus, they propose that early peaks of H3K9me3 represent nucleation sites for stable heterochromatin domains, and that piRNA-associated co-transcriptional silencing is involved in their establishment. They also identify a small number of active retrotransposon families corresponding to the earliest nucleation sites. An additional value of this study is the provided collection of H3K9me3 profiles in narrow developmental stages of *D. miranda*.

**Decision letter after peer review:**

Thank you for submitting your article "Sparse nucleation and piRNA-mediated establishment of H3K9me3-dependent heterochromatin during *Drosophila* development" for consideration by *eLife*. Your article has been reviewed by 3 peer reviewers, and the evaluation has been overseen by a Reviewing Editor and Jessica Tyler as the Senior Editor. The reviewers have opted to remain anonymous.

The reviewers have discussed the reviews with one another and the Reviewing Editor has drafted this decision to help you prepare a revised submission.

As the editors have judged that your manuscript is of interest, but as described below that additional experiments are required before it is published, we would like to draw your attention to changes in our revision policy that we have made in response to COVID-19 (https://elifesciences.org/articles/57162). First, because many researchers have temporarily lost access to the labs, we will give authors as much time as they need to submit revised manuscripts. We are also offering, if you choose, to post the manuscript to bioRxiv (if it is not already there) along with this decision letter and a formal designation that the manuscript is 'in revision at *eLife*'. Please let us know if you would like to pursue this option. (If your work is more suitable for medRxiv, you will need to post the preprint yourself, as the mechanisms for us to do so are still in development.)

Summary:

In this study, the authors characterize H3K9me3 establishment during embryonic development in *D. miranda*. The authors find that at Stage 3 there are multiple, frequently stochastic, H3K9me3 peaks. A subset of these sites, frequently proximal to transposon-targeted by piRNAs, are characterized by stable heterochromatin at Stage 7 of development. Thus, they propose that early peaks of H3K9me3 represent nucleation sites for stable heterochromatin domains, and that piRNAs are involved in their establishment.

The reviewers recognize the value of these studies particularly with respect to providing a collection of H3K9me3 profiles in narrow developmental stages. However, they also points to major issues that question the validity of the conclusions and need to be addressed for publication in *eLife*.

Essential revisions:

1) All reviewers are concerned that 'stage 3' peaks might be 'phantom peaks'. These typically appear near open regions such as promoters and enhancers even when normalized to input. This is a well-known issue (PMID:26117547), and these peaks are particularly apparent when actual peaks are sparse (such as in stage 3 embryos). A clear demonstration that 'stage 3' peaks are not phantom peaks is essential to support the validity of the authors' claims and for publication in *eLife*.

Experiments suggested by the reviewers to address this point include:

– Providing ChIP-seq data from embryos without the target of interest, in this case H3K9 histone methyltransferase mutants;

– Providing parallel H3K9me3 and "mock" IPs, e.g., with IgG, to test whether these peaks are a technical artifacts.

3) A clear characterization of embryo staging for *D. miranda* is required. In contrast to *D. melanogaster*, the biology of early embryogenesis in D.miranda is not well studied and the authors should first demonstrate by immunocytology that their staging really is correct in respect to the drawings provided in Figure 1B.

4) Limiting the ChIP-seq analysis to H3K9me3 could mask the establishment of silencing. The conclusion that so-called 'temporary H3K9me3 peaks' indicate regions not establishing heterochromatin might be a misinterpretation because it is not shown whether these are only initiated by H3K9me3 but maintained in their heterochromatic state by H3K9me2 indexing. This would be resolved by investigating H3K9me2 peaks during development.

Related to this point, the authors should show also H3K9me3 staining in salivary glands of *D. miranda*, as the two markers might look distinct.

5) No direct evidence is provided to support the claim that piRNAs has a role in heterochromatin formation. The claim is based on the observation that 'persistent' H3K9me3 peaks are proximal to piRNA-enriched TEs, but the authors should either provide direct evidence (such as by deleting or introducing piRNA-enriched TEs into the genome near some of the stage 3 peaks and see how this affects heterochromatin formation), or significantly tone down their claims, including by removing this conclusion from the title.

6) The reviewers have additional concerns about data analysis/interpretation, most of which can potentially be addressed by text changes or additional data analyses. Specifically:

a. Figure 5B: The spreading of H3K9me3 appears to follow a different pattern in different replicates. In repl. 7-8, the sharp H3K9me3 peaks remain sharp at stage7. This suggests that heterochromatin spreading does not take place around these peaks. On the contrary, repl 1-2 show that almost half of the less robust H3K9me3 peaks expand into their surrounding regions at stage7. The authors should try to explain why heterochromatin spreading is observed only in a few replicates and how do they reconcile their model with the observation that sharp peaks appear to result in less spreading. This would also influence the interpretations of Figure 6 D and Sup. 7.

b. The authors showed the patterns of stage3 H3K9me3 peaks vary depending on replicates. Does this epigenetic variegation during stage3 affect the final landscape of H3K9me3 after heterochromatin formation is completed? Although the stage3 H3K9me3 peaks are not identical between replicates, all the replicates may share the regions where the peaks are observed; in that case heterochromatin formation takes place at similar regions, which would result in very similar H3K9me3 patterns at stage7. Did the author compare the patterns of H3K9me3 at stage7 between replicates-are they identical, or different?

c. What is happening to the peaks and the surrounding regions where signals are invisible (the lower parts of each panel) in Fig6 F and G? Are the signals completely absent, or do they have weak signals with a similar tendency as observed at the peaks of the upper parts? If former is the case, how do the authors apply their model to those peaks?

d. The spike-in normalization is potentially problematic. From methods: "when there is more than one alignment of equal mapping quality, the read will be randomly assigned to one of the targets" – wouldn't this cause reads derived from real H3K9me3 peaks from *D. melanogaster* to get assigned to identical sequences of *D. miranda* due to the randomization? Are the results (especially for stage 3 peaks) reproducible without spike-ins? Will the results hold if all reads mapping to *D. melanogaster* are discarded prior to mapping to *D. miranda*?

Another reviewer notes that additional caveats might arise from using *D. melanogaster* embryos for spike-in normalization. In *D. melanogaster*, heterochromatin formation by H3K9me2 and H3K9me3 becomes visible around cycle 9-10. In stage 7 embryos gastrulation is already progressed in *D. melanogaster* and spike-in normalization with these embryos might be problematic if earlier embryos of *D. miranda* (cycle 10-13) are compared as well as by the significant differences in repeated sequences between the two species.

e. The authors note that the early non-persistent peaks are highly variable, only appearing in a few of the libraries. Two key questions arise: could the random assignment of reads to different mapping sites cause peaks and could the number of microsatellite copies in the genome be underestimated in the assembly? Random assignment of repetitive sequences to mapping sites might result in stochastic appearance of peaks, especially if there are many repeat copies and few reads. Would these peaks continue to be strong and stochastic if reads were mapped to all sites weighted by mapping site number? The authors mention that some peaks are present with unique mapping, but don't mention whether uniquely mapping peaks are similarly or less stochastic than ones that map to multiple loci. Do the authors ever observe "apparent" depletion at any of these sites compared to input (i.e., if peaks are due to random assignment there should also be "random peaks" in the input)? How certain are the authors about the exact copy number of the microsatellites? If microsatellite copies are underestimated, then peaks could appear due to "apparent" oversampling. While the authors attempted to account for this issue using DNA-seq libraries, the combination of random assignment of reads and unmapped microsatellite sequences might lead to apparent enrichments, which could appear stronger or more wide-spread due to unidentified microsatellite copies.

f. The authors argue that "technical noise does not account for the observation that peaks are enriched for microsatellites, and only a small fraction of repeats are enriched with H3K9me3 at stage 3". However, given how widespread microsatellites are across the genome (>200,000 loci), I wonder if the overlap with the stage 3 H3K9me3 is by random chance. Would random sampling of the genome with equal number of genomic regions as the stage 3 H3K9me3 TR-associated peaks result in similar overlap with TRs? If it is not random, what is special about the occupied TRs compared to the remaining ~99% of TRs in the genome – e.g., do they have higher accessibility, transcription, specific sequence?

g. While piRNA-initiated heterochromatin spreads, it is hard to imagine how a single nucleosome (143bp or 124bp genomic length) can be targeted by a TE that is hundreds or even up to 10 kb away. Spreading from this distance would result in broad peaks (like the authors observe at later stages) and not targeting of distant specific individual nucleosomes.

h. The authors should also more carefully interpret accessibility differences using ATAC-seq because significant transitions in chromatin indexing during early embryonic development in *D. melanogaster* were recently demonstrated occurring before and during heterochromatin establishment. This might also occur in *D. miranda* and could contribute to the findings reported.

7) Several citations are missing. The reviewers are particularly concerned about:

– Missing references to studies highlighting heterochromatin establishment in *D. melanogaster*, including studies on the role of H3K9me2 in gene silencing and heterochromatin formation.

– Poor citation of studies on piRNA-mediated H3K9me3 establishment.

– Missing citations of the literature of *D. miranda*, to justify why the study was done in this organism (e.g., it might be interesting to compare the model species *D. melanogaster* with the evolutionary very distant species *D. miranda* with interesting specificities like sex chromosome evolution).

8) The fact that this study as performed in *D. miranda* should be better clarified, including mentioning it in the title. The authors should also clarify what is new in this study compared to what has already been resolved in *D. melanogaster*.

[Editors' note: further revisions were suggested prior to acceptance, as described below.]

Thank you for resubmitting your work entitled "Establishment of H3K9me3-dependent heterochromatin during embryogenesis in *Drosophilamiranda*" for further consideration by *eLife*. Your revised article has been evaluated by Jessica Tyler (Senior Editor) and a Reviewing Editor.

Both reviewers agree that the study is interesting and conducted with rigor, supporting its publication. The manuscript has been improved but there are some remaining issues that need to be addressed. No additional experiments are required, but some data reanalysis might be needed. The reviewers also raise significant concerns about data presentation and interpretation, and about missing or incorrect citations, which need to be addressed before publication. Please carefully address the points raised by the reviewers in a revised version of the manuscript, with a specific attention to the comments highlighted here.

Both reviewers have issues with the authors' claims about the role of piRNAs in heterochromatin nucleation. Specifically, the concern refers to the claim that on one hand piRNAs are important for nucleation, but on the other hand piRNAs do not map at sites where the nucleations occur. The reviewers also offer somewhat different interpretation of the presented data, suggesting that he point the authors would like to make is not clear. I am copying here the comments on this point, which require significant revision of the main text, a clearer model/conclusion, and editing of the abstract.

Reviewer 2: Based on the fact that piRNA mappings along the TEs show no correlation with nucleation sites the authors speculate that some other mechanism might be responsible for nucleation (transcription itself or Zn-finger proteins) and piRNAs induce spreading. While this can certainly not be excluded, I am not sure where the expectation that a complex that binds nascent transcript would induce changes on the DNA at the same sequence as where it associates with the RNA comes from. In fact that seems relatively unlikely to me. Instead, I would think that recruitment of the complex on nascent transcript will induce nucleation in the proximity and that other factors (e.g., location of txn initiation or binding of other chromatin factors) determine where chromatin modification nucleates. Antisense piRNAs are mostly derived from piRNA clusters, which contain fragments of TEs, often lacking intact promoters (keeping all the TE promoters would increase transcriptional interference within the cluster). Thus, if nucleation occurs at promoters (as the authors observe for several TEs), piRNAs might predominantly not be able to target this region as clusters might be deprived of TSSs.

Reviewer 4: The correlation between piRNAs and H3K9me3 peaks seems marginal if at all. The last sentence of the abstract, while most likely true, seems a gross over-interpretation of the results. While I can well imagine how small RNA at one stage guide H3K9me3 at a later stage, the absence of correlation with H3K9me3 peaks make this scenario a long stretch. Therefore, the reasoning in the manuscript becomes pretty confusing. That maternally deposited piRNAs may be instrumental in suppressing this group of TEs is interesting and well possible but hard to conclude from the presented data.

On the piRNA data presentation: I don't see the value in showing correlation lines and r-values in figures 6D, E, F, 4H when there is no correlation visible. 3H, S9 (x-axis = kb from peak?) make the reader wonder how piRNA distribution look like further (than 2kb) away from the peak of H3K9me3. Is there any accumulation around peaks? Perhaps piRNA accumulation and distribution could be better grasped in browser shots like S5 and S7 with an additional piRNA track?

What is the y-axis unit in Figure 5D, G, J? For small RNAs, it is common to calculate normalized read counts in RPM (reads per total mio mapped). However, 5000RPM would be a lot?! (10,000RPM = 1% of the library) Readers need to be able to judge expression strength and depth.

Figure 6 is confusing: while it looks like lots of piRNAs map to stage 3 TEs in panel A and B, panel D (and E) shows no correlation whatsoever. Maybe it could be pointed out more clearly that A+B are about distinct TEs, while D+E are about H3K9me3 position within an element? I took the conclusion that piRNAs "mark" stage 3 high copy number TEs from panel F.

On the other hand, the authors speculate that specificity for nucleation might be provided by the act of transcription of TEs (line 396-98), which from an evolutionary arms race perspective I really don't like, as TEs like to evade repression and thus would likely change their expression profile to later stages if very early transcription initiation is the cause of more robust silencing.

Additional comments that need careful consideration, text rewriting, and potentially some re-analyses:

1) I am a little confused by the logic of how transcription is responsible for K9me3 peak initiation when the authors describe that both for stage 3 and stage 4 nucleating peaks the transcript level increases later, at stage 4 and 4-5, respectively. To me the data suggests that nucleation induces transcription (admittedly, this is going against the assumed function of K9me3) rather than the other way around.

2) Around line 308 the authors describe that the weak fold difference in K9me3 levels between the truncated and full length copies is likely a sever underestimation because non-unique mappings are distributed evenly amongst copies. Why did the authors not look at unique mappers?

3) In Figure 7B-C it is perplexing that the H3K9me3 enrichment in the last 1kb of internal sequence of TEs is higher in the full-length, while in the first 1kb internal sequence it is higher in the truncated copies. How do the authors explain this difference?

4) The H3K9me3 patterns along TEs are intriguing and very interesting. The authors may want to group and present TEs by class, at least discuss them by class in the text. This may explain "discrepancies" between transposon families. It seems LTR-retroelements show 5'-nucleation while non-LTR elements have a 3'-bias? The findings could be further elevated by discussing patterns in other organisms, perhaps data in *D. melanogaster* is available? In mouse and primates, 5'-nucleation has been observed by: Rebollo, 2011 Plos Genetics; Wolf, 2015 Genes Dev; Walter, 2016 *eLife*; Imbeault 2017 Nature; Wolf, 2020 *eLife*.

In mouse and human, this nucleation is often (but perhaps not always) guided by ZFP which the authors point out are not conserved in flies. Given the similar patterns, this may still point to a highly conserved targeting mechanism for H3K9me3.

H3K9me3 induction at several LTR-retroelements in mouse have been shown to initiate at their highly conserved tRNA primer binding site and spread from there (Rebollo, 2011 Plos Genetics; Wolf, 2015 Genes Dev). It would be helpful if the authors state more precisely what they defined as a "5'-truncation". How many nucleotides were missing in these copies? Only 5'-LTRs or perhaps additional nucleotides from the 5'-UTR?

5) Much of the RNA expression data is plotted like one would plot chromatin peaks and a bit odd presentation of "no expression". In Figure S6 and 3G (upper third coordinates) there are a couple of loci that are expressed throughout stages. What are they? Are they independent of H3K9me3 and are they a specific transposon family? Genic? The obtained RNAseq data should be more accessible, which portions of the genome are expressed at the stages sampled? Perhaps some pie charts or supplemental tables of where in the genome reads map to at which stage could provide an overview of what the RNAseq actually revealed other than negative correlation with H3K9me3 at select loci.

6) Many references are missing (especially for piRNAs and k9me nucleation) as detailed in the 'recommendations for the authors' section. The authors also identify several errors or missing text in the main text and figure, as clarified in the 'Recommendations for the authors' section that follows.

*Reviewer #2 (Recommendations for the authors):*

The resubmitted version of the manuscript makes less profound claims than the original one. This is mostly due to the fact that many of the original peaks they found in early embryos turned out to be artifacts. Based on the current results, the main claims are that nucleation at TEs starts as early as stage 3 of embryonic development and that early nucleation is predominantly on active TEs (often at their promoters) and leads to more robust H3K9 methylation at later stages. The authors find that early nucleating TEs are highly targeted by maternal piRNAs and show early zygotic transcription consistent with the prevailing model that piRNA-induced transcriptional silencing requires transcription.

The new version of the manuscript is toned down and makes claims that are consistent with the data. The authors have addressed all the concerns except for proper references, although this mainly is due to removing a large fraction of the previous data that was fund to be a technical artifact and by toning down claims that were based on correlations. Whether these current results are still sufficiently interesting for publication in *eLife* is not my responsibility to decide. I do find the observation of very early nucleation, which is consistent with piRNA-mediated early establishment of K9-methylation over TEs, interesting and, although not very surprising, still worth publishing.

I have a few remaining comments, which mostly indicate some level of sloppiness and the lack of familiarity with the piRNA field as well as some disagreement with the author's interpretations of the data.

1) I am a little confused by the logic of how transcription is responsible for K9me3 peak initiation when the authors describe that both for stage 3 and stage 4 nucleating peaks the transcript level increases later, at stage 4 and 4-5, respectively. To me the data suggests that nucleation induces transcription (admittedly, this is going against the assumed function of K9me3) rather than the other way around.

2) Based on the fact that piRNA mappings along the TEs show no correlation with nucleation sites the authors speculate that some other mechanism might be responsible for nucleation (transcription itself or Zn-finger proteins) and piRNAs induce spreading. While this can certainly not be excluded, I am not sure where the expectation that a complex that binds nascent transcript would induce changes on the DNA at the same sequence as where it associates with the RNA comes from. In fact that seems relatively unlikely to me. Instead, I would think that recruitment of the complex on nascent transcript will induce nucleation in the proximity and that other factors (e.g., location of txn initiation or binding of other chromatin factors) determine where chromatin modification nucleates. Antisense piRNAs are mostly derived from piRNA clusters, which contain fragments of TEs, often lacking intact promoters (keeping all the TE promoters would increase transcriptional interference within the cluster). Thus, if nucleation occurs at promoters (as the authors observe for several TEs), piRNAs might predominantly not be able to target this region as clusters might be deprived of TSSs.

On the other hand, the authors speculate that specificity for nucleation might be provided by the act of transcription of TEs (line 396-98), which from an evolutionary arms race perspective I really don't like, as TEs like to evade repression and thus would likely change their expression profile to later stages if very early transcription initiation is the cause of more robust silencing.

3) Around line 308 the authors describe that the weak fold difference in K9me3 levels between the truncated and full-length copies is likely a sever underestimation because non-unique mappings are distributed evenly amongst copies. Why did the authors not look at unique mappers?

4) In Figure 7B-C it is perplexing that the H3K9me3 enrichment in the last 1kb of internal sequence of TEs is higher in the full-length, while in the first 1kb internal sequence it is higher in the truncated copies. How do the authors explain this difference?

5) References remain insufficient and often wrong or at least not referring to the primary/first discovery. It is very apparent that the authors have not obtained a broad knowledge of the piRNA field and/or have done a sloppy job with the references even after this was raised in the first round of review. Here are some examples:

– Line 42: While I am not sure why the authors bring phase separation into this story, if they do, they should also be aware of counter-arguments to HP1 inducing phase separation (Erdel et al., 2020. PMID: 32101700).

– Line 51-53: The authors refer to results in flies and mice but only reference a single fly paper. Shortly after (line 57), they cite 3 papers (not correctly formatted) for Piwi causing H3K9 methylation, yet one of them, Derricarrere et al., only showed that Piwi cleavage activity is needed and did not provide any analysis of its role in H3K9 methylation, while some relevant papers are missing.

– Line 59-61: The authors say that the piRNA-Piwi complex recruits heterochromatin factors, including histone methyltransferases for H3K9me3 deposition. The papers cited refer to Piwi inducing H3K9me3, without either of them showing recruitment of the HMT. On the other hand, they miss citing the paper that comes closest to showing recruitment mechanism of the HMT. In addition, they cite one of two papers describing Panx (why not the paper by Brennecke?) and one on the SFINX complex (this time only the Brennecke paper out of 4 parallel papers), the first a factor of unknown function and the 2nd an RNA binding complex. This seems like a random choice of factors (excluding Arx, the RDC complex, etc.) and a random choice of references for those.

6) Figures and figure legends:

– Figure legend 5: panel "C" is missing.

– Color scheme in Figure 4H is missing?

– Figure 7B-D it would help if the panels would indicate what the difference is so one does not have to look in the figure legend.

– For suppl. figure 5 and 7Y axis label is missing. Are these different stages?

– For suppl. figure 11 the colors are not indicated (for the different stages, so one hast to look at the main figures to know what stage is which color). Also, the "position of called peaks" indicated at the top of panels would need to be mentioned in the figure legends in all figures.

Just a suggestion: While the figures are very nice in color (and consistent throughout the manuscript), the color choices are making it essentially impossible to look at the data on a black and white printout. While this is by itself not a huge issue, I generally prefer plots that do not depend on color printouts. E.g., in Figure 3E-F the two colors look identical in BandW and, accordingly, it is impossible to determine what the lower and upper half of the chromosome arm markings are in 3F. If the colors are left the same, it would help to better describe this in the figure legends. Numerous other examples of colors being hard to evaluate in BandW are found, that I do not list.

*Reviewer #4 (Recommendations for the authors):*

Summary:

Wei et al., determined genome-wide repressive histone H3 lysine K9 trimethylation (H3K9me3) marks during precisely timed developmental stages in *Drosophila* miranda embryos. Heterochromatin formation by H3K9me3 was profiled starting 3.5 hours post fertilization (9 cell divisions, "stage 3") up to gastrulation (14 cell divisions, "stage 7"). While a proportion of stage 3 were transient, early stage 4 H3K9me3 peaks seem to nucleate and correlate very well with repressive chromatin spreading in later stages.

The authors profile RNA expression at several stages as well as maternally deposited small RNAs right after fertilization ("stage 1"). The RNA expression data confirms silencing of loci that nucleate heterochromatin formation at stage 4. Wei et al. find abundant PIWI-interacting RNAs (piRNAs) that target transposon sequences and examine their potential to guide H3K9me3 deposition as previously observed in Drosphila *melanogaster* and other organisms. Transposons that have repressive H3K9me3 marks at stage 3 are piRNA targets, suggesting maternally deposited piRNAs may guide this early heterochromatin induction. However, piRNA target sites do not correlate with H3K9me3 positions at these loci, making the link between stage 1 piRNAs and transcriptional silencing at stage 3 rather speculative.

The detailed H3K9me3 data in *D. miranda* reveals interesting patterns of heterochromatin establishment over transposable elements (TEs). Several LTR-retrotransposons exhibit nucleation at their 5'-end, while other non-LTR elements show early enrichment at their 3'-ends. This is similar and interesting when compared to 5'-nucleation patterns of H3K9me3 during epigenetic reprogramming at TEs in mammals.

1) The H3K9me3 patterns along TEs are intriguing and very interesting. The authors may want to group and present TEs by class, at least discuss them by class in the text. This may explain "discrepancies" between transposon families. It seems LTR-retroelements show 5'-nucleation while non-LTR elements have a 3'-bias? The findings could be further elevated by discussing patterns in other organisms, perhaps data in *D. melanogaster* is available? In mouse and primates, 5'-nucleation has been observed by: Rebollo, 2011 Plos Genetics; Wolf, 2015 Genes Dev; Walter, 2016 *eLife*; Imbeault, 2017 Nature; Wolf, 2020 *eLife*.

In mouse and human, this nucleation is often (but perhaps not always) guided by ZFP which the authors point out are not conserved in flies. Given the similar patterns, this may still point to a highly conserved targeting mechanism for H3K9me3.

H3K9me3 induction at several LTR-retroelements in mouse have been shown to initiate at their highly conserved tRNA primer binding site and spread from there (Rebollo, 2011 Plos Genetics; Wolf, 2015 Genes Dev). It would be helpful if the authors state more precisely what they defined as a "5'-truncation". How many nucleotides were missing in these copies? Only 5'-LTRs or perhaps additional nucleotides from the 5'-UTR?

2) The correlation between piRNAs and H3K9me3 peaks seems marginal if at all. The last sentence of the abstract, while most likely true, seems a gross over-interpretation of the results. While I can well imagine how small RNA at one stage guide H3K9me3 at a later stage, the absence of correlation with H3K9me3 peaks make this scenario a long stretch. Therefore, the reasoning in the manuscript becomes pretty confusing. That maternally deposited piRNAs may be instrumental in suppressing this group of TEs is interesting and well possible but hard to conclude from the presented data.

On the piRNA data presentation: I don't see the value in showing correlation lines and r-values in figures 6D, E, F, 4H when there is no correlation visible. 3H, S9 (x-axis = kb from peak?) make the reader wonder how piRNA distribution look like further (than 2kb) away from the peak of H3K9me3. Is there any accumulation around peaks? Perhaps piRNA accumulation and distribution could be better grasped in browser shots like S5 and S7 with an additional piRNA track?

What is the y-axis unit in Figure 5D, G, J? For small RNAs, it is common to calculate normalized read counts in RPM (reads per total mio mapped). However, 5000RPM would be a lot?! (10,000RPM = 1% of the library) Readers need to be able to judge expression strength and depth.

Figure 6 is confusing: while it looks like lots of piRNAs map to stage 3 TEs in panel A and B, panel D (and E) shows no correlation whatsoever. Maybe it could be pointed out more clearly that A+B are about distinct TEs, while D+E are about H3K9me3 position within an element? I took the conclusion that piRNAs "mark" stage 3 high copy number TEs from panel F.

3) Much of the RNA expression data is plotted like one would plot chromatin peaks and a bit odd presentation of "no expression". In Figure S6 and 3G (upper third coordinates) there are a couple of loci that are expressed throughout stages. What are they? Are they independent of H3K9me3 and are they a specific transposon family? Genic? The obtained RNAseq data should be more accessible, which portions of the genome are expressed at the stages sampled? Perhaps some pie charts or supplemental tables of where in the genome reads map to at which stage could provide an overview of what the RNAseq actually revealed other than negative correlation with H3K9me3 at select loci.

4) The piRNA references are outdated and additional references would make the authors point only stronger.

– Line 53 and 256 (and others): There are excellent reviews on transcriptional silencing through the piRNA pathway that are much more recent than 2009, e.g. Ninova, 2019 Dev; Olovnikov, 2012 Curr Opin Genet Dev; Ozata, 2019 Nature Rev Genetics

– Line 53: Brennecke et al., 2007 was a foundational paper to characterize piRNA cluster and PIWI-mediated post-transcriptional silencing of TEs in *Drosophila* but does not show transcriptional silencing. Several other studies later did which would be more accurate to cite here, if the authors don't want to cite a review. E.g.:

*Drosophila*: Sienski et al., 2012 Cell (cited elsewhere); Le Thomas et al., 2013 Genes Dev (cited elsewhere); Huang et al., 2013 Dev Cell; Mammals: Watanabe 2011 Science; Aravin 2008 Mol Cell; Pezic, 2014 Genes Dev; *C. elegans*: Gu et al., 2012 Nature Genetics

– Line 345: If the point is metazoan, gametogenesis and preimplantation… add such references to Wang et al., 2018. E.g. Tang et al., 2016 Nat Rev Genet; Laue et al., 2019 Nature Comm (zebrafish); Ishiuchi et al., 2021 NSMB; Saitou et al., 2012 Dev (review)

---

## [Author Response]

Essential revisions:1) All reviewers are concerned that 'stage 3' peaks might be 'phantom peaks'. These typically appear near open regions such as promoters and enhancers even when normalized to input. This is a well-known issue (PMID:26117547), and these peaks are particularly apparent when actual peaks are sparse (such as in stage 3 embryos). A clear demonstration that 'stage 3' peaks are not phantom peaks is essential to support the validity of the authors' claims and for publication in eLife.Experiments suggested by the reviewers to address this point include:- providing ChIP-seq data from embryos without the target of interest, in this case H3K9 histone methyltransferase mutants;- providing parallel H3K9me3 and "mock" IPs, e.g., with IgG, to test whether these peaks are a technical artifacts.

We are glad this point was brought to our attention. As suggested, we performed additional ChIP analysis using different antibodies, and indeed, a substantial number of stage 3 peaks (especially temporary peaks) are indeed phantom peaks. We therefore have considerably re‐worked our current paper and re‐analyzed all of our data. Indeed, elimination of the former phantom peaks has allowed us to uncover nucleation sites in transposable elements that target initial H3K9me3 deposition, followed by spreading around those nucleation sites.

3) A clear characterization of embryo staging for D. miranda is required. In contrast to *D. melanogaster*, the biology of early embryogenesis in D.miranda is not well studied and the authors should first demonstrate by immunocytology that their staging really is correct in respect to the drawings provided in Figure 1B.

We have previously done careful developmental staging of *D. miranda* (and its sister species *D. pseudoobscura*) (Lott et al., PLoS Genetics 2014), which our current staging is based on. In Lott et al., our collaborators and us clearly showed that the overall developmental program is the same between *D. melanogaster* and *D. miranda* – i.e. the zygotic genome became activated at the same developmental stages as in both species. Furthermore, Kuntz and Eisen, (PLoS Genetics 2014) showed that across the *Drosophila* genus, embryonic development is highly stereotypical with consistent timing of developmental landmarks. These two papers are now cited to make this clear.

4) Limiting the ChIP-seq analysis to H3K9me3 could mask the establishment of silencing. The conclusion that so-called 'temporary H3K9me3 peaks' indicate regions not establishing heterochromatin might be a misinterpretation because it is not shown whether these are only initiated by H3K9me3 but maintained in their heterochromatic state by H3K9me2 indexing. This would be resolved by investigating H3K9me2 peaks during development.

Silencing could indeed be achieved by other means, including H3K9me2, and we made sure not to claim otherwise. Importantly, we no longer focus on temporary H3K9me3 peaks, and instead focus on how H3K9me3 is established during early embryogenesis. Note, we now show (supplementary figure 1) that H3K9me2 enrichment is almost identical to H3K9me3, albeit at later developmental stages (stage 5), but with much lower enrichment, as the H3K9me2 antibody is likely less sensitive for the purpose of ChIPseq.

Related to this point, the authors should show also H3K9me3 staining in salivary glands of D. miranda, as the two markers might look distinct.

We previously published salivary gland squashes of *D. miranda*, stained for H3K9me2 and HP1a (Zhou et al., PLoS Biol 2013; Figure 1). As expected for heterochromatin, these regions don’t polytenize and all the staining is observed in the chromocenter in females (apart from the heterochromatic island on Mulller B and E which correspond to former centromeres, and were described in previous publications; Mahajan et al., 2018 PLoS Bio, Bracewell et al., *eLife* 2019; these islands are also found in our H3K9me3 profiles; see Figure 2A). In males, the euchromatic parts of the neo‐Y show some structure and are interspersed by heterochromatin, but it is not possible to localize anything on these messy chromosome squashes (see Zhou et al., PLoS Biol 2013; or Bachtrog, MBE, 2003, Figure 4 for an attempt to map TEs on the neoY). As mentioned, we have also done H3K9me2 ChIP‐seq at embryonic stage 5, which shows highly similar enrichment pattern to H3K9me3 (supplementary figure 1).

5) No direct evidence is provided to support the claim that piRNAs has a role in heterochromatin formation. The claim is based on the observation that 'persistent' H3K9me3 peaks are proximal to piRNA-enriched TEs, but the authors should either provide direct evidence (such as by deleting or introducing piRNA-enriched TEs into the genome near some of the stage 3 peaks and see how this affects heterochromatin formation), or significantly tone down their claims, including by removing this conclusion from the title.

We tone down our claims, and also changed the title. Please note that *D. miranda* is a non‐model species, and it is currently not possible to do any of these experiments. We have extended our analyses of piRNA (sense and antisense piRNA) mapping to TEs and expression analysis, and show that early nucleating TEs are highly targeted by maternal piRNAs and show early zygotic transcription, consistent with a model of co‐transcriptional silencing of TEs by small RNAs.

6) The reviewers have additional concerns about data analysis/interpretation, most of which can potentially be addressed by text changes or additional data analyses. Specifically:a. Figure 5B: The spreading of H3K9me3 appears to follow a different pattern in different replicates. In repl. 7-8, the sharp H3K9me3 peaks remain sharp at stage7. This suggests that heterochromatin spreading does not take place around these peaks. On the contrary, repl 1-2 show that almost half of the less robust H3K9me3 peaks expand into their surrounding regions at stage7. The authors should try to explain why heterochromatin spreading is observed only in a few replicates and how do they reconcile their model with the observation that sharp peaks appear to result in less spreading. This would also influence the interpretations of Figure 6 D and Sup. 7.

We have removed this figure and the corresponding analyses, and we no longer draw conclusions based on between replicates analyses. Also, we have de‐emphasized the difference between persistent and temporary peaks.

b. The authors showed the patterns of stage3 H3K9me3 peaks vary depending on replicates. Does this epigenetic variegation during stage3 affect the final landscape of H3K9me3 after heterochromatin formation is completed? Although the stage3 H3K9me3 peaks are not identical between replicates, all the replicates may share the regions where the peaks are observed; in that case heterochromatin formation takes place at similar regions, which would result in very similar H3K9me3 patterns at stage7. Did the author compare the patterns of H3K9me3 at stage7 between replicates-are they identical, or different?

See previous reply.

c. What is happening to the peaks and the surrounding regions where signals are invisible (the lower parts of each panel) in Fig6 F and G? Are the signals completely absent, or do they have weak signals with a similar tendency as observed at the peaks of the upper parts? If former is the case, how do the authors apply their model to those peaks?

These plots/analyses are no longer employed.

d. The spike-in normalization is potentially problematic. From methods: "when there is more than one alignment of equal mapping quality, the read will be randomly assigned to one of the targets" – wouldn't this cause reads derived from real H3K9me3 peaks from *D. melanogaster* to get assigned to identical sequences of D. miranda due to the randomization? Are the results (especially for stage 3 peaks) reproducible without spike-ins? Will the results hold if all reads mapping to *D. melanogaster* are discarded prior to mapping to D. miranda?

In our mapping strategy, only regions of the genomes with 100% sequence homology will result in misassignments between the genomes. This is very rare given that the species have over 20 million years of divergence. In fact, using our strategy to map and assign a *D. miranda* male WGS sample, only 0.08% of reads are misassigned to the *D. melanogaster* genome (19,064/22,080,589 reads). Reciprocally, if we map a *D. melanogaster* male sample to the *D. miranda* genome, the incorrect assignment rate is 0.035% (14,396/40,672,211 reads). The issue with sequential mapping is that reads originating from regions with moderate homology between the species will successfully map to the other species, and we avoid this by mapping reads to a concatenated genome. Mapping the *D. miranda* male sample to the *D. melanogster* genome, we get mapping rates of 8.86% (1,957,202/22,080,589 reads), which means that these reads would be tossed regardless of the fact that they map better to the *D. miranda* genome. Even if we filter the reads by a very stringent mapping quality score (MAPQ >= 50), the mapping rate is 5.47% (1,208,224/22,080,589).

Another reviewer notes that additional caveats might arise from using *D. melanogaster* embryos for spike-in normalization. In *D. melanogaster*, heterochromatin formation by H3K9me2 and H3K9me3 becomes visible around cycle 9-10. In stage 7 embryos gastrulation is already progressed in *D. melanogaster* and spike-in normalization with these embryos might be problematic if earlier embryos of D. miranda (cycle 10-13) are compared as well as by the significant differences in repeated sequences between the two species.

In Vlassova et al., 1991, C‐banding shows differentiation of heterochromatin and euchromatin during blastoderm embryos but the nuclear cycle is not well defined. Based on Yuan and O’Farrell, (2016), H3K9me2 and H3K9me3 only becomes cytologically visible at interphase 12 and 13, respectively. Our use of a spike‐in was to ensure that the enrichment in our *D. miranda* samples can be compared and normalized to a standard. Using later stages where the H3K9me3 signal is robust is actually preferable because it provides more consistent signal of enrichment in the spike‐in. We are unsure as to why differing repeat content between the species would be problematic; in fact, it is beneficial when mapping ChIP‐seq reads, as it avoids problems with putative cross‐mapping between species.

e. The authors note that the early non-persistent peaks are highly variable, only appearing in a few of the libraries. Two key questions arise: could the random assignment of reads to different mapping sites cause peaks and could the number of microsatellite copies in the genome be underestimated in the assembly? Random assignment of repetitive sequences to mapping sites might result in stochastic appearance of peaks, especially if there are many repeat copies and few reads. Would these peaks continue to be strong and stochastic if reads were mapped to all sites weighted by mapping site number? The authors mention that some peaks are present with unique mapping, but don't mention whether uniquely mapping peaks are similarly or less stochastic than ones that map to multiple loci. Do the authors ever observe "apparent" depletion at any of these sites compared to input (i.e., if peaks are due to random assignment there should also be "random peaks" in the input)? How certain are the authors about the exact copy number of the microsatellites? If microsatellite copies are underestimated, then peaks could appear due to "apparent" oversampling. While the authors attempted to account for this issue using DNA-seq libraries, the combination of random assignment of reads and unmapped microsatellite sequences might lead to apparent enrichments, which could appear stronger or more wide-spread due to unidentified microsatellite copies.

This part has been removed from the manuscript (see our reply to 6a). Random assignment of reads at repetitive sequences will actually cause less stochasticity in peaks. For example, if there are two identical repetitive regions and only one region has enrichment, stochastic mapping will cause similar number of reads to be assigned to both copies causing two similar looking peaks. By necessity, repetitive sequences create a lot of reads which will therefore be more evenly assigned across copies by chance.

f. The authors argue that "technical noise does not account for the observation that peaks are enriched for microsatellites, and only a small fraction of repeats are enriched with H3K9me3 at stage 3". However, given how widespread microsatellites are across the genome (>200,000 loci), I wonder if the overlap with the stage 3 H3K9me3 is by random chance. Would random sampling of the genome with equal number of genomic regions as the stage 3 H3K9me3 TR-associated peaks result in similar overlap with TRs? If it is not random, what is special about the occupied TRs compared to the remaining ~99% of TRs in the genome – e.g., do they have higher accessibility, transcription, specific sequence?

This part has been removed from the manuscript (see our reply to 6a).

g. While piRNA-initiated heterochromatin spreads, it is hard to imagine how a single nucleosome (143bp or 124bp genomic length) can be targeted by a TE that is hundreds or even up to 10 kb away. Spreading from this distance would result in broad peaks (like the authors observe at later stages) and not targeting of distant specific individual nucleosomes.

The majority of the earliest nucleation sites at stage 3 (figure 3E) are found within TEs, and our paper now focuses on these.

h. The authors should also more carefully interpret accessibility differences using ATAC-seq because significant transitions in chromatin indexing during early embryonic development in *D. melanogaster* were recently demonstrated occurring before and during heterochromatin establishment. This might also occur in D. miranda and could contribute to the findings reported.

We have redone and repurposed the ATAC‐seq analyses because of these issues. We generated ATACseq from single embryos at the same developmental stages as the ChIP‐Seq. The use of single embryo here is to account for coverage differences between sex chromosomes in male and female embryos. We used the metric ATAC‐enrichment, which is number of ATAC‐seq reads divided by number of DNA‐seq reads. This accounts for the difference in read coverage of sex chromosomes and copy number in repeats. We now use ATAC‐enrichment to assess whether heterochromatic peaks and regions around it become less accessible over development.

7) Several citations are missing. The reviewers are particularly concerned about:– Missing references to studies highlighting heterochromatin establishment in *D. melanogaster*, including studies on the role of H3K9me2 in gene silencing and heterochromatin formation.– Poor citation of studies on piRNA-mediated H3K9me3 establishment.– Missing citations of the literature of D. miranda, to justify why the study was done in this organism (e.g., it might be interesting to compare the model species *D. melanogaster* with the evolutionary very distant species D. miranda with interesting specificities like sex chromosome evolution).

We have included several more references throughout the paper, and especially in the introduction, to address this concern.

8) The fact that this study as performed in D. miranda should be better clarified, including mentioning it in the title. The authors should also clarify what is new in this study compared to what has already been resolved in *D. melanogaster*.

We have changed the title, and now expanded on the benefits of the *D. miranda* genome for studying repeats and heterochromatin (page 3 and 4 in the introduction).

[Editors' note: further revisions were suggested prior to acceptance, as described below.]

Both reviewers agree that the study is interesting and conducted with rigor, supporting its publication. The manuscript has been improved but there are some remaining issues that need to be addressed. No additional experiments are required, but some data reanalysis might be needed. The reviewers also raise significant concerns about data presentation and interpretation, and about missing or incorrect citations, which need to be addressed before publication. Please carefully address the points raised by the reviewers in a revised version of the manuscript, with a specific attention to the comments highlighted here.Both reviewers have issues with the authors' claims about the role of piRNAs in heterochromatin nucleation. Specifically, the concern refers to the claim that on one hand piRNAs are important for nucleation, but on the other hand piRNAs do not map at sites where the nucleations occur. The reviewers also offer somewhat different interpretation of the presented data, suggesting that he point the authors would like to make is not clear. I am copying here the comments on this point, which require significant revision of the main text, a clearer model/conclusion, and editing of the abstract.

We agree that our wording on the role of piRNAs in heterochromatin nucleation was a bit confusing. We have reworded several sentences in the manuscript, and include new panels in Figure 3 and Figure 4, to clarify our observations and interpretations (which have been clearly spelled out by reviewer 2 as well). We find that piRNA’s are highly enriched in genomic regions containing H3K9me3 nucleation sites (Figure 3G, Figure 4H). In addition, the association between early nucleation and abundant piRNA at specific TEs (Figure 6A) as well as inferred zygotic transcription of these TEs (Figure 6C, D) is consistent with the proposed model of co‐transcriptional silencing of TEs, where PIWI/piRNA complexes bind to nascent TE transcripts and induce heterochromatin in the proximity. We de‐emphasize the fact that early H3K9me3 peaks and the positions of piRNA’s don’t overlap, since, as reviewer 2 points out as well, this may not be expected, and does not mean that piRNAs are not important in targeting heterochromatin. What we were trying to say in our manuscript is that heterochromatin nucleation happens at particular sequences on TEs (as shown in Figure 5), and this site‐specificity is not driven by piRNA. Thus, we favor a model where

PIWI/piRNAs recruit the heterochromatin machinery to nascent transcripts, and targeted nucleation at particular sequences nearby is then driven by other factors (possibly transcription initiation for TRAM TE, Figure 5, Figure 7). We are also cognizant that our results do not directly show these things, so we have toned down our wording in presenting these results in the abstract and main text.

Reviewer 2: Based on the fact that piRNA mappings along the TEs show no correlation with nucleation sites the authors speculate that some other mechanism might be responsible for nucleation (transcription itself or Zn-finger proteins) and piRNAs induce spreading. While this can certainly not be excluded, I am not sure where the expectation that a complex that binds nascent transcript would induce changes on the DNA at the same sequence as where it associates with the RNA comes from. In fact that seems relatively unlikely to me. Instead, I would think that recruitment of the complex on nascent transcript will induce nucleation in the proximity and that other factors (e.g., location of txn initiation or binding of other chromatin factors) determine where chromatin modification nucleates. Antisense piRNAs are mostly derived from piRNA clusters, which contain fragments of TEs, often lacking intact promoters (keeping all the TE promoters would increase transcriptional interference within the cluster). Thus, if nucleation occurs at promoters (as the authors observe for several TEs), piRNAs might predominantly not be able to target this region as clusters might be deprived of TSSs.

Thank you for pointing this out. We agree that it is not quite clear what the expectation should be regarding piRNA reads mapping and nucleation sites even if piRNA is the targeting mechanism to TEs. We fully agree that piRNA may direct the recruitment of HMTs to nascent transcript bringing it to the proximity of TE; nucleation at specific positions may then be signaled by other features (e.g. TSS or Znf). As mentioned above, what we were trying to say is that it is not piRNA that confer the sequence-specificity for heterochromatin formation, and we have reworded our manuscript to make this point more clear.

Reviewer 4: The correlation between piRNAs and H3K9me3 peaks seems marginal if at all. The last sentence of the abstract, while most likely true, seems a gross over-interpretation of the results. While I can well imagine how small RNA at one stage guide H3K9me3 at a later stage, the absence of correlation with H3K9me3 peaks make this scenario a long stretch. Therefore, the reasoning in the manuscript becomes pretty confusing. That maternally deposited piRNAs may be instrumental in suppressing this group of TEs is interesting and well possible but hard to conclude from the presented data.

As pointed out above, we have rephrased several statements and added new figure panels to make our point more clearly. We see several lines of evidence linking piRNA and TEs suppression: We find that genomic regions that contain early nucleating sites are enriched for piRNA’s (Figure 3G, Figure 4H). Further, we find that early nucleating TEs are highly enriched for high sense and anti‐sense maternal piRNA (Figure 6A) and that piRNA abundance is significantly correlated with the extent of H3K9me3 enrichment across TEs and this correlation becomes stronger over developmental stages (Figure 6B). This in conjunction with our finding that early nucleating TEs show early zygotic transcription (Figure 6C) led us to suggest that maternally deposited piRNAs are important for the co‐transcriptional silencing of TEs via H3K9me3 deposition, which is consistent with previously suggested models. As suggested by reviewer 2 (and we are in full agreement), these results can be interpreted as (or is consistent with the model) piRNAmediated recruitment to nascent transcripts which brings the machinery in close proximity to other features on or around TE insertions (TSS or ZNF) that direct nucleation. Nevertheless, we have toned down our conclusion/interpretation: The last sentence of the abstract now reads:

“These results support the model of piRNA‐associated co‐transcriptional silencing while also raising the possibility of additional mechanisms for site‐restricted H3K9me3 nucleation at TEs during early *Drosophila* embryogenesis.”

On the piRNA data presentation: I don't see the value in showing correlation lines and r-values in figures 6D, E, F, 4H when there is no correlation visible. 3H, S9 (x-axis = kb from peak?) make the reader wonder how piRNA distribution look like further (than 2kb) away from the peak of H3K9me3. Is there any accumulation around peaks? Perhaps piRNA accumulation and distribution could be better grasped in browser shots like S5 and S7 with an additional piRNA track?

For figures 3H, S9, 6D, we show the regression lines and Pearson’s correlation coefficients precisely to show the lack of correlation. However, for figures 6E and F, we do not agree that the correlations are of no value. For 6F, there is a visually clear and strongly statistically significant negative correlation between piRNA abundance and TE transcript abundance; this is visible even without the regression line. For 6E, we show increasing correlation between mean enrichment across TE and mean piRNA mapping to the TE. The associations here are visibly noisy but the correlations are strong for the later developmental stages (4e and 7) as per the p‐values and correlation coefficients.

We also added piRNA distribution of up to 10kb around the peaks in Figure 3 —figure supplement 4 and Figure 4 —figure supplement 3B, which do not show piRNA enrichment around peaks (or specific positions near peaks), but as pointed out above, genomic regions containing H3K9me3 peaks are strongly enriched for piRNA (Figure 3G, Figure 4H). Tracks of sense and antisense piRNA mapping within TEs are found in Figure 5D,G and J.

What is the y-axis unit in Figure 5D, G, J? For small RNAs, it is common to calculate normalized read counts in RPM (reads per total mio mapped). However, 5000RPM would be a lot?! (10,000RPM = 1% of the library) Readers need to be able to judge expression strength and depth.

It is (was) the piRNA read coverage across the TE. It is now converted to RPM.

Figure 6 is confusing: while it looks like lots of piRNAs map to stage 3 TEs in panel A and B, panel D (and E) shows no correlation whatsoever. Maybe it could be pointed out more clearly that A+B are about distinct TEs, while D+E are about H3K9me3 position within an element? I took the conclusion that piRNAs "mark" stage 3 high copy number TEs from panel F.

We have made the distinction between 6A, D and F more clear in the figure and the figure legends.

Indeed, 6D (now E) is looking at piRNA mapping and H3K9me3 at specific positions of TEs while the rest are for aggregate and averaged abundance of piRNA/enrichment/transcript abundance across the entire TE.

On the other hand, the authors speculate that specificity for nucleation might be provided by the act of transcription of TEs (line 396-98), which from an evolutionary arms race perspective I really don't like, as TEs like to evade repression and thus would likely change their expression profile to later stages if very early transcription initiation is the cause of more robust silencing.

For a TE to transpose, it needs to be transcribed. We are not trying to say that early transcription per se is responsible for silencing, but the current model of co‐transcriptional silencing of TEs invokes nascent transcript for piRNAs to target their genomic region. We are showing that TEs that are early transcribed are exactly the ones that are silenced the earliest (TEs with stage 3 peaks also show zygotic transcription between stage 2 and 4). For the TEs that nucleate after stage3 (stage4 TEs in figure 6), they show increase in expression between stage 4 and stage 5 (Figure 6C). We interpret this to mean that even if a TE like TRAM evolve to become late expressing, the same mechanism for transcription‐associated nucleation is likely also present. These results are consistent with the current model for cotranscriptional silencing of TEs where piRNA and PIWI target TE nascent transcripts (Ozata et al., 2019). From an arms‐race stand point, we would actually speculate that early expressing TEs may have more time to remain highly expressed prior to full heterochromatin establishment, despite having early nucleation.

Additional comments that need careful consideration, text rewriting, and potentially some re-analyses:1) I am a little confused by the logic of how transcription is responsible for K9me3 peak initiation when the authors describe that both for stage 3 and stage 4 nucleating peaks the transcript level increases later, at stage 4 and 4-5, respectively. To me the data suggests that nucleation induces transcription (admittedly, this is going against the assumed function of K9me3) rather than the other way around.

The model of co‐transcriptional silencing of TEs involves targeting of the PIWI/piRNA complex to nascent transcripts, which in turn recruits the silencing machinery, including HMT, that induce H3K9me3. Thus, transcription is in fact necessary for heterochromatin formation, and has been experimentally demonstrated in other organisms (but, as far as we know, not yet in *Drosophila*).

2) Around line 308 the authors describe that the weak fold difference in K9me3 levels between the truncated and full length copies is likely a sever underestimation because non-unique mappings are distributed evenly amongst copies. Why did the authors not look at unique mappers?

Unfortunately, if we use only unique‐mapping reads, less than 5% of the full-length insertions show uniquely‐mapping reads through the body of the TE.

3) In Figure 7B-C it is perplexing that the H3K9me3 enrichment in the last 1kb of internal sequence of TEs is higher in the full-length, while in the first 1kb internal sequence it is higher in the truncated copies. How do the authors explain this difference?

For these boxplots, we are averaging across only parts of the elements in both the full length and (most of the) truncated elements; this is meant to provide comparison between homologous regions between the truncated and full‐length copies. For 7B (5’truncations), we are averaging across 1000‐2708bp of the internal coding sequence of both of the full‐length insertions. However, for 7C (3’ truncations), we are averaging across 1‐1708bp of full‐length elements. Because of this, the enrichment in 7C includes the 5’ nucleation (elevated H3K9me3) sites thus elevating the averaged enrichment, while those in 7B lack the 5’ nucleation sites thus lowering the averaged enrichment.

4) The H3K9me3 patterns along TEs are intriguing and very interesting. The authors may want to group and present TEs by class, at least discuss them by class in the text. This may explain "discrepancies" between transposon families. It seems LTR-retroelements show 5'-nucleation while non-LTR elements have a 3'-bias? The findings could be further elevated by discussing patterns in other organisms, perhaps data in *D. melanogaster* is available? In mouse and primates, 5'-nucleation has been observed by: Rebollo, 2011 Plos Genetics; Wolf, 2015 Genes Dev; Walter, 2016 eLife; Imbeault 2017 Nature; Wolf, 2020 eLife.

We have now grouped TEs by their classes in Figure 5A. We also grouped LTR retrotransposons with 5’ nucleation in supplementary figure 10 and non‐LTR retrotransposons in supplementary figure 11. While some LTRs (TRAM, BEL‐5 and Gypsy‐11) show clear 5’ nucleation, we are unsure whether this can be generalized to all LTRs. For the non‐LTR retrotransposons, we see nucleation in various positions. The R1 variant in figure 5 show nucleation near 3’, however another variant in Supp figure 11 shows nucleation near 5’. The LOA‐2 element has nucleation near the 5’ but is still ~1kb away. We have added an observed 5’ bias in mice in the discussion.

In mouse and human, this nucleation is often (but perhaps not always) guided by ZFP which the authors point out are not conserved in flies. Given the similar patterns, this may still point to a highly conserved targeting mechanism for H3K9me3.

True. This is why we talk about ZAD‐ZNF in *Drosophila* in the discussion.

H3K9me3 induction at several LTR-retroelements in mouse have been shown to initiate at their highly conserved tRNA primer binding site and spread from there (Rebollo, 2011 Plos Genetics; Wolf, 2015 Genes Dev). It would be helpful if the authors state more precisely what they defined as a "5'-truncation". How many nucleotides were missing in these copies? Only 5'-LTRs or perhaps additional nucleotides from the 5'-UTR?

Insertions with 5’ and 3’ truncations are those that have the respective LTRs removed and at least has 500bp of the remainder of the internal sequence. We have now added Figure 7 – supplement figure 1C showing the lengths and structures of the truncated copies.

5) Much of the RNA expression data is plotted like one would plot chromatin peaks and a bit odd presentation of "no expression". In Figure S6 and 3G (upper third coordinates) there are a couple of loci that are expressed throughout stages. What are they? Are they independent of H3K9me3 and are they a specific transposon family? Genic? The obtained RNAseq data should be more accessible, which portions of the genome are expressed at the stages sampled? Perhaps some pie charts or supplemental tables of where in the genome reads map to at which stage could provide an overview of what the RNAseq actually revealed other than negative correlation with H3K9me3 at select loci.

We added Figure 3 – supplement 3, which shows a heatmap of the transcript abundance (in TPM) of genes that overlap with stage 3 peaks across development. 40% of these genes are maternally deposited and appear as those streaks in the figures mentioned (Figure 3G and Figure 3 – supplement figure 2 which was previously S6). Figure 3 – supplement 3B shows the breakdown (in pie chart) of the proportion of the genes overlapping stage 3 peaks that have maternal transcripts compared to that of all genes. Consistent with our conclusion that maternal transcripts of these genes are unlikely to contribute to early stage 3 nucleation, no difference (40% vs 39.8%) was observed. We also now added a supplementary file 2 documenting the mapping statistics (to TEs, genes) of the RNA‐seq data, and adjusted the color scheme of the heatmaps to make them more clear.

6) Many references are missing (especially for piRNAs and k9me nucleation) as detailed in the 'recommendations for the authors' section. The authors also identify several errors or missing text in the main text and figure, as clarified in the 'Recommendations for the authors' section that follows.

We have added all the recommended literature (see below).

Reviewer #2 (Recommendations for the authors):The resubmitted version of the manuscript makes less profound claims than the original one. This is mostly due to the fact that many of the original peaks they found in early embryos turned out to be artifacts. Based on the current results, the main claims are that nucleation at TEs starts as early as stage 3 of embryonic development and that early nucleation is predominantly on active TEs (often at their promoters) and leads to more robust H3K9 methylation at later stages. The authors find that early nucleating TEs are highly targeted by maternal piRNAs and show early zygotic transcription consistent with the prevailing model that piRNA-induced transcriptional silencing requires transcription.The new version of the manuscript is toned down and makes claims that are consistent with the data. The authors have addressed all the concerns except for proper references, although this mainly is due to removing a large fraction of the previous data that was fund to be a technical artifact and by toning down claims that were based on correlations. Whether these current results are still sufficiently interesting for publication in eLife is not my responsibility to decide. I do find the observation of very early nucleation, which is consistent with piRNA-mediated early establishment of K9-methylation over TEs, interesting and, although not very surprising, still worth publishing.I have a few remaining comments, which mostly indicate some level of sloppiness and the lack of familiarity with the piRNA field as well as some disagreement with the author's interpretations of the data.1) I am a little confused by the logic of how transcription is responsible for K9me3 peak initiation when the authors describe that both for stage 3 and stage 4 nucleating peaks the transcript level increases later, at stage 4 and 4-5, respectively. To me the data suggests that nucleation induces transcription (admittedly, this is going against the assumed function of K9me3) rather than the other way around.

Addressed above.

2) Based on the fact that piRNA mappings along the TEs show no correlation with nucleation sites the authors speculate that some other mechanism might be responsible for nucleation (transcription itself or Zn-finger proteins) and piRNAs induce spreading. While this can certainly not be excluded, I am not sure where the expectation that a complex that binds nascent transcript would induce changes on the DNA at the same sequence as where it associates with the RNA comes from. In fact that seems relatively unlikely to me. Instead, I would think that recruitment of the complex on nascent transcript will induce nucleation in the proximity and that other factors (e.g., location of txn initiation or binding of other chromatin factors) determine where chromatin modification nucleates. Antisense piRNAs are mostly derived from piRNA clusters, which contain fragments of TEs, often lacking intact promoters (keeping all the TE promoters would increase transcriptional interference within the cluster). Thus, if nucleation occurs at promoters (as the authors observe for several TEs), piRNAs might predominantly not be able to target this region as clusters might be deprived of TSSs.

Addressed above.

On the other hand, the authors speculate that specificity for nucleation might be provided by the act of transcription of TEs (line 396-98), which from an evolutionary arms race perspective I really don't like, as TEs like to evade repression and thus would likely change their expression profile to later stages if very early transcription initiation is the cause of more robust silencing.

Addressed above.

3) Around line 308 the authors describe that the weak fold difference in K9me3 levels between the truncated and full-length copies is likely a sever underestimation because non-unique mappings are distributed evenly amongst copies. Why did the authors not look at unique mappers?

Addressed above.

4) In Figure 7B-C it is perplexing that the H3K9me3 enrichment in the last 1kb of internal sequence of TEs is higher in the full-length, while in the first 1kb internal sequence it is higher in the truncated copies. How do the authors explain this difference?

Addressed above.

5) References remain insufficient and often wrong or at least not referring to the primary/first discovery. It is very apparent that the authors have not obtained a broad knowledge of the piRNA field and/or have done a sloppy job with the references even after this was raised in the first round of review. Here are some examples:– Line 42: While I am not sure why the authors bring phase separation into this story, if they do, they should also be aware of counter-arguments to HP1 inducing phase separation (Erdel et al., 2020. PMID: 32101700).

We include the citation.

– Line 51-53: The authors refer to results in flies and mice but only reference a single fly paper. Shortly after (line 57), they cite 3 papers (not correctly formatted) for Piwi causing H3K9 methylation, yet one of them, Derricarrere et al., only showed that Piwi cleavage activity is needed and did not provide any analysis of its role in H3K9 methylation, while some relevant papers are missing.

We include more citations. Citation of Derricarrere et al., was meant for “Independent of the posttranscriptional cleavage activity” at the beginning of the sentence. We have moved the citation to right after it.

– Line 59-61: The authors say that the piRNA-Piwi complex recruits heterochromatin factors, including histone methyltransferases for H3K9me3 deposition. The papers cited refer to Piwi inducing H3K9me3, without either of them showing recruitment of the HMT. On the other hand, they miss citing the paper that comes closest to showing recruitment mechanism of the HMT. In addition, they cite one of two papers describing Panx (why not the paper by Brennecke?) and one on the SFINX complex (this time only the Brennecke paper out of 4 parallel papers), the first a factor of unknown function and the 2nd an RNA binding complex. This seems like a random choice of factors (excluding Arx, the RDC complex, etc.) and a random choice of references for those.

We include more citations.

6) Figures and figure legends:– Figure legend 5: panel "C" is missing.

Added.

– Color scheme in Figure 4H is missing?

Added.

– Figure 7B-D it would help if the panels would indicate what the difference is so one does not have to look in the figure legend.

Added.

– For suppl. figure 5 and 7Y axis label is missing. Are these different stages?

Y‐axes and stage labels are now added.

– For suppl. figure 11 the colors are not indicated (for the different stages, so one hast to look at the main figures to know what stage is which color). Also, the "position of called peaks" indicated at the top of panels would need to be mentioned in the figure legends in all figures.

Added color legends and positions of called stage 3 peaks are now indicated by gray boxes.

Just a suggestion: While the figures are very nice in color (and consistent throughout the manuscript), the color choices are making it essentially impossible to look at the data on a black and white printout. While this is by itself not a huge issue, I generally prefer plots that do not depend on color printouts. E.g., in Figure 3E-F the two colors look identical in BandW and, accordingly, it is impossible to determine what the lower and upper half of the chromosome arm markings are in 3F. If the colors are left the same, it would help to better describe this in the figure legends. Numerous other examples of colors being hard to evaluate in BandW are found, that I do not list.

Unfortunately given the number of stages to compare using other means of discriminating them (like dotted lines or different shaped points) just end up being even harder to interpret. We changed the colors in 3E and F for better discrimination. For F the two colors are actually plotted separately – one on top and one on the bottom.

Reviewer #4 (Recommendations for the authors):1) The H3K9me3 patterns along TEs are intriguing and very interesting. The authors may want to group and present TEs by class, at least discuss them by class in the text. This may explain "discrepancies" between transposon families. It seems LTR-retroelements show 5'-nucleation while non-LTR elements have a 3'-bias? The findings could be further elevated by discussing patterns in other organisms, perhaps data in *D. melanogaster* is available? In mouse and primates, 5'-nucleation has been observed by: Rebollo, 2011 Plos Genetics; Wolf, 2015 Genes Dev; Walter, 2016 eLife; Imbeault, 2017 Nature; Wolf, 2020 eLife.In mouse and human, this nucleation is often (but perhaps not always) guided by ZFP which the authors point out are not conserved in flies. Given the similar patterns, this may still point to a highly conserved targeting mechanism for H3K9me3.H3K9me3 induction at several LTR-retroelements in mouse have been shown to initiate at their highly conserved tRNA primer binding site and spread from there (Rebollo, 2011 Plos Genetics; Wolf, 2015 Genes Dev). It would be helpful if the authors state more precisely what they defined as a "5'-truncation". How many nucleotides were missing in these copies? Only 5'-LTRs or perhaps additional nucleotides from the 5'-UTR?

Addressed above.

2) The correlation between piRNAs and H3K9me3 peaks seems marginal if at all. The last sentence of the abstract, while most likely true, seems a gross over-interpretation of the results. While I can well imagine how small RNA at one stage guide H3K9me3 at a later stage, the absence of correlation with H3K9me3 peaks make this scenario a long stretch. Therefore, the reasoning in the manuscript becomes pretty confusing. That maternally deposited piRNAs may be instrumental in suppressing this group of TEs is interesting and well possible but hard to conclude from the presented data.On the piRNA data presentation: I don't see the value in showing correlation lines and r-values in figures 6D, E, F, 4H when there is no correlation visible. 3H, S9 (x-axis = kb from peak?) make the reader wonder how piRNA distribution look like further (than 2kb) away from the peak of H3K9me3. Is there any accumulation arounds peaks? Perhaps piRNA accumulation and distribution could be better grasped in browser shots like S5 and S7 with an additional piRNA track?

Addressed above.

What is the y-axis unit in Figure 5D, G, J? For small RNAs, it is common to calculate normalized read counts in RPM (reads per total mio mapped). However, 5000RPM would be a lot?! (10,000RPM = 1% of the library) Readers need to be able to judge expression strength and depth.

Addressed above.

Figure 6 is confusing: while it looks like lots of piRNAs map to stage 3 TEs in panel A and B, panel D (and E) shows no correlation whatsoever. Maybe it could be pointed out more clearly that A+B are about distinct TEs, while D+E are about H3K9me3 position within an element? I took the conclusion that piRNAs "mark" stage 3 high copy number TEs from panel F.

Addressed above.

3) Much of the RNA expression data is plotted like one would plot chromatin peaks and a bit odd presentation of "no expression". In Figure S6 and 3G (upper third coordinates) there are a couple of loci that are expressed throughout stages. What are they? Are they independent of H3K9me3 and are they a specific transposon family? Genic? The obtained RNAseq data should be more accessible, which portions of the genome are expressed at the stages sampled? Perhaps some pie charts or supplemental tables of where in the genome reads map to at which stage could provide an overview of what the RNAseq actually revealed other than negative correlation with H3K9me3 at select loci.

Addressed above.

4) The piRNA references are outdated and additional references would make the authors point only stronger.– Line 53 and 256 (and others): There are excellent reviews on transcriptional silencing through the piRNA pathway that are much more recent than 2009, e.g. Ninova, 2019 Dev; Olovnikov, 2012 Curr Opin Genet Dev; Ozata, 2019 Nature Rev Genetics.

References added.

– Line 53: Brennecke et al., 2007 was a foundational paper to characterize piRNA cluster and PIWI-mediated post-transcriptional silencing of TEs in *Drosophila* but does not show transcriptional silencing. Several other studies later did which would be more accurate to cite here, if the authors don't want to cite a review. E.g.:Drosophila: Sienski et al., 2012 Cell (cited elsewhere); Le Thomas et al., 2013 Genes Dev (cited elsewhere); Huang et al., 2013 Dev Cell; Mammals: Watanabe 2011 Science; Aravin 2008 Mol Cell; Pezic, 2014 Genes Dev; *C. elegans*: Gu et al., 2012 Nature Genetics

We added several references.

– Line 345: If the point is metazoan, gametogenesis and preimplantation… add such references to Wang et al., 2018. E.g. Tang et al., 2016 Nat Rev Genet; Laue et al., 2019 Nature Comm (zebrafish); Ishiuchi et al., 2021 NSMB; Saitou et al., 2012 Dev (review)

References added.